

# GPR signature of Quaternary faulting: a study from the Mt. Pollino
# region, southern Apennines, Italy.
Maurizio Ercoli[1-4], Daniele Cirillo[2-4], Cristina Pauselli[1-4], Harry M. Jol[3], Francesco Brozzetti[2-4]
[1]: *Università degli Studi di Perugia, Dipartimento di Fisica e Geologia, Piazza dell'Università 1, 06123 Perugia,*
*Italy.*
[2]: *Universita "G. d'Annunzio" di Chieti-Pescara, DiSPUTer, via dei Vestini 31, 66100 Chieti, Italy.*
[3]: *University of Wisconsin - Eau Claire, Department of Geography and Anthropology, 105 Garfield Avenue, Eau*
*Claire, WI, 54702.*
[4]: *CRUST Centro inteRUniversitario per l'analisi SismoTettonica tridimensionale, Italy.*
*Correspondence to*: Maurizio Ercoli (maurizio.ercoli@unipg.it; maurizio.ercoli@gmail.com)
**Abstract.** With the aim of unveiling evidence of Late Quaternary faulting, a series of Ground Penetrating Radar (GPR)
profiles were acquired across the Campotenese continental basin (Mt. Pollino region) in the southern Apennines active
extensional belt (Italy). A set of forty-nine 300 MHz and 500 MHz GPR profiles, traced nearly perpendicular to a
buried normal fault, were acquired and carefully processed through a customized workflow. The data interpretation
allowed us to reconstruct a pseudo-3D model depicting the boundary between the Mesozoic bedrock and the
sedimentary fill of the basin, which were in close proximity to the fault. Once reviewing and defining the GPR
signature of faulting, we highlight in our data how near surface alluvial and colluvial sediments appear to be dislocated
by a set of conjugate (west and east-dipping) discontinuities that penetrate inside the underlying Triassic dolostones.
Close to the contact between the continental deposits and the bedrock, some buried scarps which offset wedge-shaped
deposits are interpreted as coseismic ruptures, subsequently sealed by later deposits. Although the use of pseudo-3D
GPR data implies more complexity linking the geophysical features among the radar images, we have reconstructed
a reliable subsurface fault pattern, discriminating master faults and a series of secondary splays. We believe our
contribution provides an improvement in the characterization of active faults in the study area which falls within the
Pollino seismic gap and is considered potentially prone to severe surface faulting. Our aim is for our approach and
workflow to be of inspiration for further studies in the region as well as for similar high seismic hazard areas
characterized by scarcity of near-surface data.
***Key-words:*** *Ground Penetrating Radar (GPR); Image processing; Faults; Neotectonics; Palaeoseismology;*
*Earthquake hazards.*
## 1. Introduction
A *"seismic gap"* is a region characterized by active faults that are seismically *"silent"* during the historical record
(Galadini and Galli 2003), but which are considered, on the basis of paleoseismological and/or morpho-structural data,
able to generate significant earthquakes (McCann et al. 1979; Mogi 1979; Plafker and Galloway 1989). Although the
hypothesis stating that the earthquake potential increases after a long quiet period was rejected by some authors (Kagan
and Jackson 1991), in some cases fault segments located in areas of longstanding gaps have revealed their seismic
potential, by producing earthquakes of moderate or high magnitude. Late Quaternary activity related to normal faults
can be suggested by structural and geomorphological evidence, as happens also in many sectors of the active Italian



extensional province, but their seismogenic role of such faults is not well understood yet. As shown by the 2016-2017
seismic sequence in central Apennines (Chiaraluce et al. 2017; Galli et al. 2019), the Italian Apennines can be
considered an ideal laboratory for all geoscience disciplines studying seismic gaps. This seismic crisis occurred in an
area displaced by an alignment of west-dipping Quaternary faults, where no important earthquakes had been recorded
over the past 1500 years. Nine-strong and moderate seismic events struck a ~80 km long region, climaxing in the Mw
= 6.5 "Norcia" mainshock (Porreca et al. 2018 and references therein; Ercoli et al. 2020), producing extensive surface
faulting (Pucci et al. 2017; Civico et al. 2018; Villani et al. 2018; Brozzetti et al. 2019, 2020; Testa et al. 2019; Cirillo
2020). Geological and paleoseismological studies across the faults set were preceding this seismic sequence (Calamita
et al. 1992; Brozzetti and Lavecchia 1994; Barchi et al. 2000; Galadini and Galli 2003, Galli et al. 2008). Besides,
some years before this sequence, 2D/3D GPR surveys (Ercoli et al. 2013a; 2014) detected a peculiar geophysical
signature confirming buried past surface faulting in Holocene deposits, that was not very prominent on the present
topography. These studies revealed occurrence of strong paleo-earthquakes and led to consider the Mt. Vettore master
fault "silent", but prone to cause similar strong events in the future. Thise proves the paleoseismological approach can
be of great help in many tectonically active areas of the Apennines. Its systematic application may provide answers
on both the seismic potential and the recurrence times (Mc Calpin; Galli at al. 2008; 2019; 2020). But on many active
faults, contrary to geophysical surveys, paleoseismological data cannot be carried out systematically due complex
logistics, environmental restrictions, the relatively high costs and, authorization processes. In addition, in many
terrestrial areas, past coseismic ruptures can be poorly visible at surface, as undergoing to natural or/and anthropogenic
topographic "regularization". Thus, when also other geological, seismological, and geophysical data are scarce,
outdated, or not detailed enough for the definition of a satisfactory seismotectonic framework, Quaternary silent faults
in "*seismic gaps*" deserve great attention for local seismic hazard evaluation . For the above reasons, Quaternary faults
and associated basins have been and are currently investigated not only Central Italy, but also in the southernmost
Apennines sector (Galli et al. 2008; 2020).  A dedicated research project (Agreement INGV-DPC 2012-2013 and
2014-2015, Project S1 - Base-knowledge improvement for assessing the seismogenic potential of Italy, Brozzetti et
al. 2015; Pauselli et al. 2015) aiming to improve the base-knowledge of seismogenic structures was focused during
the 2012-2015 period also on the Calabrian region (Southern Italy).
In the project, structural geological, geophysical, and paleoseismological studies were successfully acquired across
the Mt. Pollino and Castrovillari fault systems (northern Calabria), supporting evidence of their Late Quaternary
activity (Ercoli et al. 2013b; Cinti et al. 2015, Ercoli et al. 2015; Brozzetti et al., 2017b). This area, which is considered
one of the most important seismic gaps in southern Italy, extends northward to include the Mercure and Campotenese
basins, both characterized by Late Quaternary continental syn-tectonic sedimentation (Fig. 1a-c).
The fresh geomorphic signature of the faults bounding these basins suggests Late Quaternary activity; besides,
paleoseismological trenching and radiocarbon dating document the occurrence of $M_w > 6.5$ paleo-earthquakes (Cinti
et al. 1997, 2002; Michetti et al. 1997, 2000). But the data contrasts with the poor historical seismicity, reporting a
single significant $M_w = 5.6$ event occurred in 1693 (Guidoboni et al. 2018) and, ultimately, also with instrumental
seismicity characterized by two moderate seismic sequences respectively climaxed in the Mw 5.6 Mercure (1998,
September 9) and Mw 5.2 Mormanno (2012, October 25) earthquakes. The latter occurred during a long-lasting
sequence spanning the period 2010-2014, which included more than 6000 seismic events of $M_w > 1$ and activated at
least three individual seismogenic sources (Passarelli et al., 2015; Brozzetti et al., 2017a; Cirillo et al., 2021; Fig. 1b).
The gap between the low energy release, observed during the instrumented seismic sequences, and the high seismic
potential estimated for the Quaternary faults, raised the question of whether even stronger earthquakes had shaken and



could shake the area in the future. A recent and detailed parameterization of the Fosso della Valle-Campotenese fault
(VCT), that bounds a homonym basin (Fig. 1c) showing both geological and seismological evidence of current activity
(Totaro et al. 2014, 2015; Brozzetti et al. 2017a, Cirillo et., 2021), assesses a surface length of ~15 km and a depth
of at least 10 km (Brozzetti et al. 2017a; Cirillo et al., 2021): the potential rupture-area is likely estimated to produce
$M_w > 6.0$ earthquakes. Such seismic events, in the Apennines extensional belt, are generally considered capable of
producing coseismic surface breaks (Pantosti and Valenzise 1990; Cello et al. 2000; Vittori et al. 2000; Boncio et al.
2010; Villani et al. 2018; Brozzetti et al. 2019). However, evidence of Quaternary faulting for this structure is currently
unclear, but geological and morpho-structural data suggest this fault has played an important role in determining the
geometry and the recent sedimentary evolution of the basin.
The Campotenese basin and its VCT boundary fault is an emblematic example that summarizes the aforementioned
issues: 1) unavailability of paleoseismological data as the basin is entirely located within the Mt. Pollino National
Park, thus requiring prior authorization from competent authorities; 2) unavailability of publically geophysical data;
3) no fresh surface displacements of Holocene deposits, suggesting the occurrence of recent strong earthquakes, have
been observed so far along its trace.
For all these reasons, the VCT represents an ideal case study suitable to test our working method, as it we have
conducted an explorative GPR campaign as a screening tool for the detections of potential unknown Quaternary co-
seismic ruptures along its fault zone. It is possible that, during past strong earthquakes, propagation of coseismic
ruptures along the fault could have displaced Holocene deposits at the surface, being subsequently buried by further
and more recent alluvial deposits or erased by the anthropogenic and agricultural activity. The objectives of the project
are to: i) review and describe geophysical characteristics associated with a peculiar GPR signature of faulting, and
propose a methodological GPR workflow for the study site which might be extended to other areas; ii) check the
efficiency of a GPR prospection in the Campotenese study site to locate the trace of the studied fault as well as
verifying its spatial continuity at depth; iii) provide new data to relate to the occurrence of $M_w > 6.0$ events on the
VCT, as observed for the nearby Castrovillari faults; iv) highlight new elements for more exhaustive characterization
of the VCT with the purpose of better quantifying its seismogenic potential; v) pave the way for other local geophysical
prospections and identify interesting sites for future ground-truthing and/or paleoseismological trenching; vi) to have
direct application and impact to the planning of future mitigation strategies for the reduction of surface faulting risk
in the nearby urbanized areas.
**2. Tectonic setting and seismicity**
The Campotenese continental basin is located in the northernmost Calabria region south-west of the Mt. Pollino
calcareous massif (southern Italy, Fig. 1). The bedrock of the basin consists of shallow water dolostones and
limestones, Late Triassic to Middle Miocene in age, belonging to the Verbicaro tectonic unit (Ogniben 1969; Amodio
Morelli et al. 1976). It is generally referred to the western edge of the "Apenninic Platform", a thick (>4 km) carbonate
shelf, that underwent compression during the Middle-Late Miocene times and was translated over an eastern basinal
domain (Lagronegro-Molise basin; Patacca and Scandone 2007; Vezzani et al. 2010 and references therein). From the
bottom to the top, the bedrock succession includes late Triassic dolostones, Cretaceous limestones, and Paleocenic-
Lower Miocenic calcarenites cross-cut by the pillow lava basalts belonging to Liguride units of the northern sector of
Calabrian arc (Quitzow 1935, Grandjaquet and Grandjaquet 1962, Amodio Morelli et al. 1976, Ghisetti and Vezzani,
1983; Iannace et al. 2004, 2005 and 2007; Liberi et al. 2006; Filice et al. 2015 and Tangari et al. 2018).





The origin of the Campotenese basin, however, is related to a set of NW-SE striking extensional faults which, during
the Middle-Late Pleistocene, displaced the contractional tectonic pile, favoring the deposition of alluvial and lacustrine
sediments in a subsiding intra-mountain depression (Servizio Geologico d'Italia 1970). This set of conjugate SW- and
NE-dipping normal faults represents the local expression of the Quaternary extensional belt that develops all along
the Italian peninsula, nearly parallel to the axial zone of the Apennines, from northern Tuscany to the Calabrian Arc
(Brozzetti 2011). North of Campotenese, (Lucania and southern Campania) the Apennine extensional belt includes
several continental basins and their boundary faults, as the Irpinia, Vallo di Diano, Tanagro, Melandro-Pergola and
Val d'Agri (Ascione et al. 1992; Maschio et al. 2005; Amicucci et al. 2008; Villani and Pierdominici 2010; Brozzetti
2011; Filice and Seeber 2019 and Bello et al. 2021). To the south, it continues with the Crati graben that dissects the
northern sector of the Calabrian Arc (Tortorici et al. 1995; Brozzetti et al. 2017b).
On the regional scale, the Quaternary normal fault array controls the release of major seismicity, as suggested by the
distribution of supra-crustal instrumental earthquakes (INGV 2020 and Iside 2007) and of the strongest historical
events (Fig. 1a, Tertulliani and Cucci 2014; Rovida et al. 2020). Several seismological and paleo-seismological
investigations claim that most of the faults bounding the Quaternary basins are seismogenic and therefore enable, in
some cases, to associate major past earthquakes with specific structures (Pantosti and Valenzise 1990; Cello et al.
2003; Galli et al. 2006; Spina et al. 2008; Brozzetti et al. 2009; Villani and Pierdominici 2010; Brozzetti et al. 2017a).
These same studies highlight that the kinematics of the Quaternary faults and the focal mechanisms of the major
earthquakes are mutually consistent and are compatible with an SW-NE direction of extension (RCMT and TDMT
databases by Pondrelli, 2002 and Scogliamiglio et al. 2006).
The area investigated in this work and the structures bounding the Campotenese basin have been pointed out recently
in more detail by Brozzetti et al. (2017a), in the frame of a larger study of the Quaternary and active fault pattern on
the Calabria-Lucania border (Fig. 1b,c). In the region, three main sets of normal faults have been mapped: a western
one, consisting of east- to NNE-dipping faults (red lines in Fig. 1b), and two other main sets of W-to SW-dipping fault
segments (blue lines in Fig.1b). The Campotenese Fault (VCT) is the westernmost splay of a SW-dipping fault referred
to as Rotonda - Campotenese fault, which develops in a N160E average strike-direction and includes several right-
stepping en-echelon segments (Fig. 1b). The VCT extends from the southern border of the Mercure basin to the SW
boundary of the Campotenese basin (average strike ~N155E) for ~15 km. Across its northern segment, an associated
throw of ~120 m has been assessed based on the displaced stratigraphic boundaries mapped within the bedrock
(passage between Triassic dolostones and Jurassic limestones). In this same sector, prevailing dip-slip kinematics has
been documented by Brozzetti et al. (2017a). Along the east side of the Campotenese basin, the VCT is generally
buried by Holocene deposits, but its localization can be inferred based on stratigraphic observations and geomorphic
features, such as sharp ridge fronts, linear scarps, and slope breaks. The VCT controls the distribution and thickness
of the clastic fill basin (Middle Pleistocene-Holocene in age, according to Schiattarella et al. 1994) that reaches the
maximum thickness (> 30 m) in the western sector (VCT hanging wall, see boreholes stratigraphy at
http://sgi2.isprambiente.it/mapviewer/) whereas is very thin (generally < 2-3 m) in the eastern one.  The spatial
relationships, at surface and depth, between the Quaternary fault segments, and the hypocenters of the re-located
events of the 2010-2014 seismic activity (Totaro et al. 2015; Brozzetti et al. 2017a; Napolitano et al. 2020, 2021;
Pastori et al. 2021) suggest that the VCT is a good candidate as a seismogenic source for the Mw 5.2 (2012, October
25) Mormanno mainshock. In addition, the analysis of the historical seismicity highlights that the epicenter of the Mw
5.5, 1708 earthquake (Rovida et al. 2020) is located within the VCT hanging wall block, close to its northern
termination, leading to hypothesize the possibility of a common seismogenic source with the Mormanno 2012 event.





Ultimately, based on the available data (e.g. Brozzetti et al. 2017a), the VCT must be considered an active and
seismogenic fault, whose size (length of surface trace and depth of detachment) provides an estimate of seismogenic
potential much greater than that released over the last few centuries, thus capable of giving surface faulting in the
future. Following this line of reasoning, it is hypothesized that some strong paleo-earthquakes, unknown to date, may
have caused, along the VCT trace, coseismic surface breaks that are currently no longer visible as they have been
leveled by erosion and anthropogenic activity. The search for evidence of such possible paleo-ruptures is therefore a
fundamental first step to define the actual seismogenic potential of the VCT and perhaps, to try filling in the Pollino
seismic gap.

168                                   FIGURE 1 HERE

**3. Methodology**
Ground Penetrating Radar (GPR) is a high-resolution geophysical method able to provide detailed images of the
shallow sub-surface. This methodology is based on the recording of Em echoes, with operative frequencies for
geoscience applications generally between 10 MHz and 1000 MHz, depending on the transmitting and receiving
antennae. The GPR reflections rise from dielectric permittivity contrasts between the subsurface targets and the
surrounding media, which in geological and archaeological applications typically correspond to geo-lithological
changes or water content variations (Jol 2009). In "low-loss" materials (Davis and Annan 1989), the investigation
depth range is generally within the first ten meters or less. The latter is however controlled also by the electrical
conductivity, which for high values causes radar signal attenuation (Annan 2001). The reflections are recorded as a
function of the Two-Way-Travel time (TWT) propagation, and displayed as a 1D GPR trace. Several GPR traces
displayed along a transect build-up a radar profile or "radargram", that is the 2D representation of the GPR reflections,
more commonly identified as the conventional GPR output. A GPR dataset may be provided also as a 3D volume,
which has been common for 25+ years in research applications and recently more widespread due to a wider diffusion
of commercial GPR instruments equipped with arrays of antennae. The GPR is used in many research and applied
fields, such as geological, sedimentological, hydrogeological applications (Bristow and Jol. 2003; Jol 2009), and also
in archaeological and engineering studies (Conyers 2006, 2016; Daniels 2004; Goodman and Piro 2013; Utsi 2017).
Several 2D/3D GPR studies have already imaged buried tectonic structures. These studies have shown geophysical
images of faulting, supporting and/or extending outcrop, borehole, trench data, and contributing to base-knowledge
of seismogenic structures as well as to the seismic hazard assessment of several regions around the world. Among the
pioneers, we can mention Benson (1995), Smith and Jol (1995), Busby and Merritt (1999), Cai et al. (1996) and Liner
and Liner (1997), and on the successive twenty years, other 2D GPR studies were achieved across several faults
(Audru et al. 2001; Demanet et al. 2001; Overgaard and Jakobsen, 2001; Bano et al. 2002; Liberty et al. 2003; Reiss
et al. 2003; Slater and Niemi, 2003; Malik et al. 2007; Wallace et al. 2010; Yalciner et al. 2013; Imposa et al. 2015;
Anchuela et al. 2016; Nobes et al. 2016; Matos et al. 2017; Pousse-Beltran et al. 2018; Zajc et al. 2018; Zhang et al.
2019 and Shaikh et al. 2020). A few GPR surveys have been acquired across Italian normal faults (Salvi et al. 2003;
Jewell and Bristow, 2006; Pauselli et al. 2010; Roberts et al. 2010; Ercoli et al. 2013a; Bubeck et al. 2015; Cinti et
al. 2015). Over time, such 2D GPR studies were flanked by an increasing number of pseudo-3D or full-3D GPR studies
(Grasmueck et al. 2005). Grasmueck and Green (1996) traced the future path of three-dimensional GPR applications,
providing a dense 3D GPR volume to image fractures in a Swiss quarry. The study opened the possibility to three-
dimensional GPR imaging of subsurface geological structures. Successive studies extended the approach to





characterize active faults in different tectonic regimes combining 2D and pseudo-3D GPR surveys (Gross et al. 2000,
2002, 2003, 2004; Green et al. 2003; Horstmeyer et al. 2005; Tronicke et al. 2006; McClymont et al. 2008, 2009,
2010; Vanneste et al. 2008; Christie et al. 2009; Carpentier et al. 2012a,b; Malik et al. 2012; Brandes et al. 2018). A
review of the near-surface GPR faulting studies suggests some reflection characteristics as possible indicators for the
detection of subsurface fractures and faults (e.g. Smith and Jol 1995; Liner and Liner 1997; Reiss et al. 2003; Gross
et al. 2004; McClymont et al. 2008, 2010 and Bubeck et al. 2015). Among these, sharp lateral reflectivity variations,
interruptions of the reflections, and the presence of hyperbolic diffractions are considered convincing evidence, as
shown also by numerical simulations (Ercoli et al. 2013a; Bricheva et al. 2021). In addition, we have accounted for
additional GPR indicators identified for Quaternary faulting in similar environments (Ercoli et al. 2013a,b; 2014;
2015), which are linked to the geometry of stratigraphic deposits across fault zones: i) reflections abrupt truncating
and offsetting along sub-vertical discontinuities (especially in the case of a normal fault); ii) reflection packages
thickening as they approach the fault strands; iii) abrupt lateral dip variation of the reflections; iv) peculiar reflection
package geometries, with contorted reflection patterns resembling *"colluvial wedges"*, which McCalpin (2009)
defines as deposit due to "*subsidence and sedimentation of the hangingwall and erosion of the morphological scarp*
*in the footwall*"; v) localized strong GPR signal attenuation due to the presence of conductive media within the main
fault zone (possibly associated with colluvial wedges).
Based on the research and criteria reviewed above, we carried out our interpretation of near-surface faulting based on
the co-existence of most of these features along several adjacents analyzed GPR profiles. These conditions strengthen
the interpretation of each profile and aids to highlight the spatial continuity of the interpreted structures over linear
distances of at least many tens, or hundreds, of meters.
***3.1 GPR and GNSS survey***
The GPR profiles were acquired across the VCT fault (Fig. 1c), during three different geophysical campaigns in the
years 2014-2015 (Fig. 2). The entire dataset encompasses 49 radar profiles (linear length of about 4100 m) collected
with a Common Offset (CO) configuration.

223                              FIGURE 2 HERE

We used a Zond 12e GPR system equipped with 300 and 500 MHz antennae. The lower frequency antennae was
ultimately preferred and considered the best trade-off between maximum resolution and achievable signal penetration
(in our case ~ 4 m) concerning the surveyed materials and wanted subsurface structures. The GPR was equipped with
an odometer wheel to measure the radar profiles' length and with a Topcon GR-5 Global Navigation Satellite System
(GNSS) receiver to achieve accurate positioning of GPR traces and profile. Considering the scarce presence of
obstacles across the survey site and the good satellite coverage, we opted for a Network Real-Time Kinematic
positioning (NRTK, connected to the NETGEO network), measuring coordinates and elevations with centimetre
accuracy, and stored directly within the SEG-Y GPR files.
Three datasets were acquired after preliminary fieldwork and collection of geological structural data at the surface and
which allowed us to infer the possible location of the fault trace. The average NE-SW direction of the GPR lines was
initially planned with the primary purpose of intersecting the VCT fault ~perpendicularly to its SW-NE strike, as
reported by literature and visible by surface evidence. This solution theoretically allows a more reliable interpretation



of the investigated structure by reducing the effect of the apparent dip-direction and dip-angle of both stratifications
and faults.
The first acquisition carried out in 2014, resulted in twelve SW-NE GPR profiles collected in the southern sector
(CMT light-blue lines in Fig. 2a), which was a flat land characterized by Quaternary alluvium. The second acquisition
encompassed four additional radar profiles collected in the same area, and another nine radar profiles progressively
moving to north, which were collected with slightly different and converging orientations in the central sector (CMT
green lines Fig. 2a). This solution was pursued for two main reasons: 1) to avoid directly surveying the outcropping
dolostones (only partially crossed with two northernmost profiles) characterizing two hills *h1* and *h2* (dashed white
polygons in Fig. 2), thus focussing only on the sedimentary cover which is our target for possible Quaternary faulting;
2) to optimize, using subsurface data, the future acquisition schemes by figuring out the effective "apparent dip" of
the geologic structures, to consider in the interpretation of 2D GPR images (similar to the interpretation of reflection
seismic profiles).
In order to intercept several possible buried faults and fault-related structures as well as to fully image the local
structural setting, the successive 2015 acquisition crossed part of the Triassic dolostones ridge with longer GPR
profiles. Preliminary results shown by GPR profiles collected during the second 2014 campaign (close to *h1* and *h2*)
revealed a considerable difference in GPR reflectivity between the unconsolidated deposits and layered and fractured
Mesozoic lithotypes (Gafarov et al. 2018). Therefore, two new datasets of 24 GPR parallel radar profiles (CMT dark-
blue sets of lines in Fig. 2a, *north "n"* and *south "s"*) were extended in NNE-SSW and NE-SW directions,
respectively, crossing *h1* for several tens of meters (max profile length ~220 m) throughout the basin. The GPR profiles
were recorded using a trace distance of 0.05 m and a profile inter-distance of 10 m for dataset *"n"* and 25 m for dataset
*"s"*, respectively. A detailed summary of the acquisition parameters used in the field during the GPR surveys is
reported in Table I. For these two new grids, the profile spacing and positioning are more regular and accurate, thanks
to a preparatory transects planning using a Geographic Information System Information System (GIS) project. Thus,
we later staked out their initial and final positions during the fieldwork through the differential Global Navigation
Satellite System (GNSS). The results of the accurate GPR traces positioning achieved during the GNSS campaigns
were later used also for GPR data processing, visualization, and interpretation.

262                                              TABLE 1 HERE

### 3.2 GPR data processing and results:
The processing sequence was customized after testing several flows and parameters. We aimed to remove random and
coherent (e.g. ringing) noise and enhance the data quality to better visualize the geometry of the buried reflections and
their discontinuity in signal amplitude and phase. The first step was an accurate Quality Control (QC) of the profile
coordinates and topographic profiles. Although the favorable environmental conditions of the site for a GNSS survey,
some measurements were occasionally suffered a degradation of positional accuracy (e.g. temporary scarce satellite
coverage or poor communication via Network Transport of RTCM via Internet Protocol - Ntrip). For some traces
therefore the coordinates and elevation field records that were outliers were corrected using various strategies (e.g.
replacement, interpolation, or smoothing, Figs. 3a,b).

272                                              FIGURE 3 HERE





We have also compared our measurements with topographic transects extracted from a 10 m and a 5 m resolution
Digital Terrain Models (DTM) by Tarquini et al. (2012) and by Regione Calabria. Later on, we finally used a 1 m
resolution DTM (Geoportale Nazionale, Lidar data provided by Italian Ministero dell'Ambiente e della Tutela del
Territorio e del Mare - MATTM) to double-check if, despite the different scales of observation, the topographic
profiles were comparable. Although the metre resolution of the DTM is unable to represent centimetre topographic
variations, the comparison confirmed an excellent match of the topographic profiles at a meter scale, so that the DTM
data were integrated to correct the GNSS measured topography when the accuracy of GNSS recordings were
excessively degraded. With the topographic profiles corrected, the raw GPR data (Fig. 3c, illustrating the profile
cmt5s) were initially processed with the Prism software (Radar System, Inc., http://www.radsys.lv/en/index/) using a
basic processing sequence, to analyze the main characteristics of data and optimize a customized processing flow. The
processing sequence was later improved through ReflexW software (https://www.sandmeier-geo.de/reflexw.html, see
Table II for details on the processing algorithms and parameters). The workflow included a time-zero correction,
dewow, amplitude recovery, velocity analysis, background removal, bandpass filtering, F-K filtering, 2D time
migration, topographic correction, and time-to-depth conversion. The amplitude recovery was operated through a
"gain function" including by a linear and an exponential coefficient $(g(t)=(1+a*t)*e^{(b*t)})$ to enhance the amplitude
(reflectivity) contrasts as well as preserving the horizontal and vertical amplitude variations already visible in the raw
data (Fig. 3a). This amplitude recovery function was used across all the profiles with slight customization depending
on the datasets (details in table II). The entire processing flow was applied to all the available radar profiles, again
with occasional filtering adaptations aiming to remove local pervasive signal ringing (e.g. due to low antennae-ground
coupling). Particular care was dedicated to the migration process, whose algorithm was decided after extensive tests
on several radar profiles to select the best migration strategy.
TABLE 2 HERE
In fact, a very different reflectivity and maximum depth of penetration are visible in the data: it is more than 150 ns
in the central sector, reducing to 70-80 ns in the rest of the radar profiles (Fig. 3c): this fact suggests sharp lateral
variations of subsurface media (Figs. 3d) and possibly of the velocity field. Thus, we have first tested a 1D time
migration algorithm (Kirchhoff) performing a Migration Velocity Scan (MVS) analysis (Forte and Pipan 2017) and
inspecting the success of diffraction hyperbola collapse after migration. We have varied constant values of Em
velocity, from a minimum of 0.06 up to 0.12 m/ns, with steps of 0.01 m/ns, to evaluate considerable variation in
dielectric properties of surveyed media. The MVS highlighted a higher velocity for the central sector of the GPR
profiles which displays high reflectivity: Fig. 4 illustrates an extract of the migration results obtained on the profile
cmt1n_a, by using three constant values of average velocity. The profile in Fig. 4a shows the unmigrated version
characterized by numerous hyperbolic and half hyperbolic diffractions originated by single scatter points and wavy
reflections (white arrows). In Fig. 4b we display the first test using v = 0.07 m/ns, showing overall good results, with
slightly under-migration at a few points mainly located within the shallower sediments (light-blue arrows). The
hyperbolic diffractions are also nicely collapsed using higher velocity (v = 0.09 m/ns) as shown in Fig. 4c (dark-blue
arrows), even if some imaging problems are rising on deeper reflections. The last migration scan test (v =0.11 m/ns)
displays a good result only in few profile sectors (dark-blue arrows), particularly localized within the sectors with high
reflectivity, displaying an improved lateral reflection continuity. The rest of the radar profile shows general poor





imaging, particularly in the area characterized by strong attenuation, where the wavy reflection is clearly over-
migrated (red arrows indicating migration smiles, Fig. 4d).

313                                   FIGURE 4 HERE

The workflow, therefore, suggests a challenging imaging task, due to velocity variation happening not only in depth
as well as laterally across the different media. This sharp change of reflectivity and velocity at a distance of about 13-
14 m (Fig. 4d) represents a complex problem for the efficiency of 1D migration algorithms standardly used for GPR
imaging. Such considerations have driven to test a 2D migration algorithm, by creating and using a 2D velocity model
obtained for each radar profile through a hyperbolic diffraction fitting tool (Fig. 5a). Single velocity points have been
fitted for each area clearly displaying hyperbolic diffractions, while in the remaining parts of the radar profiles we
have arbitrarily included presumed velocity adaptation only to obtain a regular grid of points to spatially interpolate
the 2D models. The 2D migrated radar profiles, in comparison to the 1D approach, resulted in improved imaging of
GPR profiles, displaying a more accurate collapse of the hyperbolic diffractions into point sources and an improved
relocation of dipping reflections, with a refinement of their geometry and an increase of their continuity. A good-
quality imaging result is visible on the central sectors of radar profiles displaying strong reflectivity and reflections
with improved continuity, but also many phase breaks and displacements. Despite steep topographic gradients, sharp
lateral velocity variation and the reflection heterogeneity might cause imaging issues to be treated using more specific
workflows (Lehmann and Green, 2000; Heincke et al. 2006; Goodman et al. 2007;  Dujardin and Bano 2013), we
believe we have reached a good compromise for our purposes. In our case, a considerable improvement, can be seen
along the hill-slope and flatter areas (profile cmt1n_a, Fig. 5b) which are of greatest interest for the study aimed at
detecting possible earthquake ruptures within the Quaternary deposits. The improved imaging of reflection geometries
is therefore fundamental for the interpretation and detection of geophysical signatures of faults.

332                                   FIGURE 5 HERE

A successive import of the processed SEG-Y was done into the seismic interpretation software OpendTect Pro v.6.4
(Academic license courtesy of dGB Earth Science, https://www.dgbes.com), which was used first for global quality
control of processing operations (correctness of topographic correction and datum plane, coordinates accuracy and
matching, profiles orientation and intersection) and for three-dimensional (3D) visualization of all the profiles (Fig.
6a).
The three-dimensional GPR project was subsequently integrated with geological and structural maps, DTM, and
literature schemes (using a common Coordinate Reference System: WGS84 UTM Zone 33N, EPSG: 32633) in the
Move suite software v. 2019.1 (Academic license courtesy of Petroleum Experts, https://www.petex.com/) for the
GPR interpretation and model building. All the east and west-dipping fault surfaces were created interpolating the
fault-sticks picked on displaced reflections and correlated across adjacent radar profiles. In particular, we used the
"*surface geometry*" tool to extract the properties of each single mesh building up the surfaces, and obtaining the "dip"
and "dip azimuth" data. Subsequently, such values have been automatically saved in an attribute table, which can then
be queried to reconstruct the "synthetic" stereonets.
**4. GPR data description and interpretation**



The 3D MOVE project allowed us to extract 2D and 3D visualizations of the radar profiles acquired across the VCT
fault trace and allows us to better figure out the relationships between the main reflections identified on the different
GPR data (Fig. 6a). The workflow aimed to reconstruct and model the three-dimensional surfaces including horizons
and high-angle discontinuities.

351                                                FIGURE 6 HERE

A common feature on all the radar profiles is the strong reflectivity visible within their central sectors (profile cmt3n,
Fig. 6b), which are characterized by a more irregular and steeper slope, particularly within the northern portion of the
surveyed area. These sectors with deep penetration areas are caused by the Triassic dolostones, which outcrop in the
central and northern portions of the study area (Figs. 1c and 2a). In the southern side of *h1*, thin microbialitic laminae
allows one to measure the attitude of the bedding, which shows a NNW dip (~ 30-35°). In the same area, we measured
two sets of major and minor joints, both with a dip angle of ~ 40-45 degrees and a SW and SE dip, respectively.
Looking at the quality of the radar reflections and at the remarkable depth (~ 6 m, Fig. 6b) reached by the GPR signal,
this rock type represents an excellent dielectric medium (corresponding to higher frequency content zone in the 2D
spectrum of Fig. 6c). However, its reflection pattern is not spatially homogenous, being often characterized by oblique
and sub-parallel reflections interpretable as dolostone beds, displaying moderate (25-30°) west and east "apparent"
dip on the respective sides of the surveyed dolostone hills. In addition, these reflections are frequently cut and slightly
displaced by apparent high-angle (60-65°) phase discontinuities, also highlighted by a dense hyperbolic diffractions
pattern (radar profile cmt2n, Fig. 7a), interpreted as sets of joints or minor faults fracturing and displacing the
dolostone layers (Fig. 7b). Apart from its internal heterogeneities, the GPR signature of the Triassic dolostones can be
considered as a well-defined depositional facies (*fc1*) (Sangree and Widmier 1979; Huggenberger 1993; Beres et al.
1999; Jol and Bristow 2003). This radar signature was recorded not only in correspondence of the outcropping
carbonates but also in the transition slope areas, where just a thin soil layer or a scarce sedimentary cover was present
onsite (Figs. 7b,c). A different GPR facies (*fc2*) is characterized by laterally-continuous and sub-parallel prominent
reflections in the very shallow depth, just beneath the direct arrivals (< 1 m), which stratigraphically seals the
underlying reflections; more discontinuous, wavy and contorted reflections of moderate to low reflectivity are visible
in the 1-3 m depth-range (variable across the analyzed profiles) onlapping onto a generally prominent and wavy
reflection (Figs. 7a,b). Below this reflection pattern (~ 2-3 m), the signal strongly attenuates. Summarizing , the
shallower reflections packages show continuous beds dipping parallel to the slope, but under these, the slightly deeper
reflections are less continuous, displaying variable dip and locally a contorted pattern, with numerous diffraction
hyperbolas (in unmigrated data, Fig. 7a) as well as important lateral amplitude variations (Fig. 7b).

377                                                FIGURE 7 HERE

We have classified the overall radar signature of these reflection packages as facies *fc2*, corresponding to the
alluvial/colluvial deposits (Fig. 7b,c,d) outcropping on the flatten sectors, which represent the GPR profile sectors
we've carefully inspected to find for geophysical evidence of Quaternary faulting. A key-layer for this research is in
our opinion the described wavy reflection, recognized in several radar profiles. This prominent reflection shows
frequently a stepped geometry, with frequent breaks of its continuity and lateral depth variations; moderate to strong
signal attenuation generally underlies this reflection. The related interpretation is not straightforward in the absence
of direct data (e.g. boreholes and/or paleoseismological trenches) or at least without additional geophysical data. A



strong reflection suggests significant variation of a dielectric constant between the two media so that most of the
incident energy is reflected back to the receiver at the surface, which is potentially explained by several geological
models, such as: i) a high dielectric contrast may be a result of a sharp soil moisture variation (Ercoli et al. 2018); ii)
a sharp erosional, stratigraphic or tectonic boundary within heterogeneous deposits (Ercoli et al. 2015), or iii) a contact
between two considerably different lithologies, such as unconsolidated deposits laying above a bedrock substrate (e.g.,
Frigeri and Ercoli 2020) reflecting back all (or almost all) the incident signal. In addition, the possible role of
conductive sediments within layered deposits (e.g. high clay content) should not be discounted, and it might explain
localized strong signal attenuation.
To support the GPR data interpretation, some suggestions for consideration are: 1) stratigraphy of two water-wells
located only ~2.5 km away on the north-westernmost sector of the Campotenese basin (Brozzetti et al. 2017a); 2)
geomorphological and geological data achieved through aerial-photo interpretation and field study on the surrounding
landscape (Brozzetti et al. 2017a); 3) the geometrical characteristics shown by this reflection and of the underlying
reflection pattern visible in the processed data.
The available well logs show the Pleistocene-Holocene alluvium and colluvium layered above the carbonate bedrock
~20-30 m depth (Brozzetti et al. 2017a), a greater depth than the strong GPR reflection. However, it should be observed
that the drilled area is located over the depocenter of the basin whereas the studied GPR site is placed just on its eastern
border, in proximity to emerged calcareous hills. In addition, it should be mentioned that only terraced Middle-
Pleistocene silts and sands (Schiattarella et al. 1994) and slight coatings of Late Pleistocene colluvium (generally < 2
m thick) are documented to outcrop in the eastern sector of the basin (footwall of VCT fault) (see Fig. 7 in Brozzetti
et al. 2017a).
The subsurface geometries highlighted by the GPR profiles suggest a relatively thin layer of sedimentary deposits
resting on a fractured substratum, whose top surface is progressively deepening towards the west, thus providing
increased space for settling sediments. For this reason, a gradual deposits thickening is observed from east to west.
Therefore, we interpret the prominent and wavy GPR reflection as a buried top layer of carbonate (e.g. as observed
by Bubeck et al. 2015), in our case the Upper Triassic dolostones formation lying beneath shallow and poorly
consolidated Quaternary deposits. Thus, in this surveyed sector of the basin, we interpret the dolostone top to be
located not ~20-30 m deep as in the noth-westernmost area, but at a shallower depth (1-3 m) below the topographic
surface and across both two sides of the surveyed hills. After picking this strong reflection in all the radar profiles, we
have therefore reconstructed the top of bedrock surface (Fig. 8a).
We can now focus on the structural interpretation by analyzing the geophysical characteristics of this strong reflection.
As illustrated by a recent structural map of the basin in Fig. 8a (modified after Brozzetti et al. 2017a), the area is
dissected by a set of en-echelon fault splays connected to the VTC master fault. The strong reflection interpreted as
the dolostone top shows a clear "stepped" geometry (Figs. 5b-6b-7b-8b), highlighting abrupt lateral variations in depth
and in its thickness (sediment growth and onlaps). We also notice other geophysical features, which can be observed
in the stratigraphy of overlying deposits fc1: some reflections are semi-continuous to discontinous (sharp variation in
signal amplitude and phase), displaying lateral dip variation.

421                          FIGURE 8 HERE

In some sectors these broken reflection packages present truncantions (Smith and Jol 1995), vertical offset, and
hyperbolic diffraction events. Contorted reflections across the main discontinuities frequently show localized strong





attenuation of GPR signal (interpretation summarized in caption of Fig. 8b), which might be linked to their high dip-
angle, causing a minor amount of energy being reflected back to the antenna, but more likely, due to the presence of
conductive fine soils across faulted zones (e.g. circles 1 and 2 in Fig. 8b). These conditions can be linked to different
depositional facies across fault zones (McClymont et al. 2010) e.g. including colluvial wedges (Reiss et al. 2013;
Bubeck et al. 2015). Such peculiar GPR signature is therefore compatible with possible co-seismic displacement due
to Late Quaternary surface faulting events (Fig. 8b). Using such stratigraphic evidence and geophysical markers of
faulting, we have therefore interpreted and classified synthetic (west-dipping, blue) and antithetic (east-dipping, red)
normal faulting events within the Quaternary sediments (Fig. 8b). The interpreted faults present a dip angle between
65-75° and a vertical offset ranging from a few tens of centimeters up to ~1 meter, and were picked using solid lines
(fault sticks). During the interpretation process, when the presence of geophysical markers of faulting were uncertain,
a dashed fault segment has been initially added and only later analyzed a second time by looking at their possible
connection with nearby lines. In addition to the fault sticks within the Quaternary sectors, we placed main fault sticks
across the sharp boundary between the Triassic dolostone and the Quaternary sediments. The contact is generally
characterized by fractured zones including hyperbolic diffractions (in unmigrated data), contorted reflections
geometry of variable dip angle, abrupt truncations marked by sharp lateral variation of the reflectivity due to localized
attenuation (Figs. 3 to 9).
By interpolation of the fault sticks placed in adjacent profiles, we have created the fault surfaces that show a good
degree of continuity, from north to south (Fig. 9).
Our geophysical interpretation allowed reconstructing the fault-network and the geometry of the associated
synsedimentary deposits, at a higher resolution (Fig. 9). The 3D map views of Fig. 9 show a structural scheme of the
main fault lineaments displaying a NW-SE strike. Our interpretation highlights an en-echelon system of main SW and
NE-dipping faults on both sides of the hills where the Triassic dolostones crop out, as well as several secondary
structures within the Quaternary sediments.
**5. Discussion**
**5.1 Inferences from subsurface 3D model**
The high-angle GPR discontinuities identified in the study, dissecting not only Quaternary alluvial-colluvial deposits
but also deeper reflections referred to deeper stratigraphic layers, show a considerable continuity in the NW-SE
direction (Fig. 9). These structures can be interpreted as the surface expression of an articulated set of extensional
meso-faults associated with the VCT.
The reconstructed faults are arranged in a horst-graben structure in which the higher displacements are associated to
the west-dipping faults (Fig. 9). Among these faults, the master structure in the southern sector of the basin (Fig. 9a)
bound toward west a structural high where the Triassic dolostones crop out. The structural high, which corresponds
to a topographic high elongated in the NNW-SSE direction, is also bordered towards the east by a minor east-dipping
normal fault (Fig. 9b). Thus, it appears as an uplifted horst showing an axis of elongation sub-parallel to the average
strike of the Campotenese basin. The horst is mostly buried within the basin but locally emerges from the Quaternary
sediments, as in the central portion of the investigated area (*h1* and *h2* in Fig. 2a).
The fault-set d1 in Fig. 9c shows the maximum displacements and the most evident deformations of the adjacent sub-
surface deposits. It can be considered a splay of the VCT (Fig. 8b), separated by a right step-over of about 0.5 km
from the main segment that borders the eastern basin (Figs. 2c, 8a).



The three-dimensional model (Fig. 9a,c) highlights that these faults, despite having an average NW-SE direction, are
characterized by a complex polymodal pattern of strikes, with NS sections alternating to NW-SE sections. The
character, which is observed at various scales, both along the entire VCT and along all the extensional structures of
the area (Brozzetti et al. 2017a), is also confirmed by the statistical analysis of the reconstructed fault planes reported
in the stereo plots of Fig. 9d (d1-d3= west-dipping faults; d2-d4= east-dipping faults).
FIGURE 9 HERE
The variations of thickness of the Quaternary deposits, detectable on the radar profiles, are consistent with the horst
and graben configuration. Thinning is observed in correspondence with the raised buried blocks whereas thickening
and wedge-shaped and locally chaotic geometries occur in correspondence with the lowered blocks. These latter
features are only associated with west-dipping faults, thus enhancing the role of this fault set (Fig. 8b), and the syn-
tectonic nature of the Late Quaternary sedimentation.
**5.2 Seismic hazard implications**
In many cases, seismological data show that the outcropping Quaternary faults are capable of releasing earthquakes,
but the maximum expected magnitude is not well constrained. An estimate of can be made using well-known scale-
relationships (Wells and Coppersmith 1994; Wesnousky et al. 2008, Leonard 2010, Stirling et al. 2013) with
knowledge of the geometric parameters (e.g. fault length, area and depth), which are often difficult to assess.
These scale relations can be applied also on scarps originated by cumulated coseismic surface faulting events of
medium-strong earthquakes (generally M > 6), possibly distinguishing the amount of slip due to each event through
paleoseismological analysis. But in the case of VCT, no direct information is available on the nature of the surveyed
deposits, and their accurate dating has not been carried out at the present day. The interpretation of our GPR data
confirms the segmentation of the VCT and point out the presence of a buried splay, which appears to have exerted a
strong control on the deposition of Late Quaternary sediments, just below the present topographic surface. Moreover,
we have highlighted that the sediments, in turn, are affected by faulting. Their location at a very low depth (1 - 4 m)
in a flat land of an intra-mountain basin which is presently undergoing alluvial and colluvial sedimentation, suggests
their attribution to the Holocene. Thus, pointing out normal faulting of Holocene deposits would be, in itself, a very
important result for the Campotenese area, as no previous study has provided this kind of evidence yet. In fact, a
Middle-Late Pleistocene age of activity was suggested for the Mercure and Campotenese boundary faults by
Schiattarella et al. (1994) and Brozzetti et al. (2017a), based on morpho-structural observations, whereas their
Holocene activity had been only inferred according to the possible associations of the faults with the recent 2010-2014
Pollino seismic sequence (Brozzetti et al. 2017a).
Our data seem even more promising because the GPR facies interpretation highlights the possible presence of small-
scale grabens or half-graben and fault-related coseismic-wedges (e.g. as observed by Reiss et al. 2013 and Bubeck et
al. 2015), at shallow depth (just below the present agricultural soil). This inference would testify to not only the
persistence of extensional deformations up to the very Late Quaternary but would even imply the occurrence of
episodes of surface faulting, proving new perspectives on the actual seismic hazard of the area. In other words, the
Campotenese basin may have been affected in the relatively recent past by medium-strong earthquakes, nucleated
from the VCT, capable of producing surface coseismic scarps, which were subsequently erased by footwall erosion
and sedimentation at the hanging wall.





The hypothesis of past earthquakes, with a magnitude sufficiently high to cause surface ruptures in this area, sounds
reasonable if we consider that historical events with $6 < M_w < 7$ are documented a little further (~50 km) north and
south (Fig. 1a) (Guidoboni et al. 2018).
Some paleoseismological earthquakes with inferred magnitude up to $M_w = 7$ are attributed to the Castrovillari fault,
located at about 20 km to SE and also falling within the Pollino seismic gap (Cinti et al. 1997; Michetti et al. 1997;
Cinti et al. 2002, 2015; and Ercoli et al. 2015).
The estimates of the VCT fault-length (Brozzetti et al. 2017a) provide an overall value of 15 km which is compatible
with the maximum expected magnitudes capable to produce surface breaks. From using earthquake scaling
relationships based on coseismic rupture-length, as the main computation parameter, the obtained magnitudes are in
the range of 6.45 (Wells and Coppersmith 1994) to 6.8 (Wesnousky et al. 2008; Leonard 2010) in the case of a
complete rupture of the VCT fault. The result suggests that the most recent earthquakes that have affected the study
area (2012 - $M_w$ 5.2; 1894 - $M_w$ 5.1; 1708 - $M_w$ 5.8 and perhaps 1693 - $M_w$ 5.6) have a source at ~ 8 km depth (Totaro
et al. 2015; Brozzetti et al. 2017a; Napolitano et al. 2020 and 2021, Sketsiou et al., 2021). Thus, the seismic energy
released is likely too low for an active fault set as well as not enough to be causative of the buried VCT ruptures
detected. Because historical catalogs do not show events with $M_w > 6$ (Guidoboni et al. 2018), probably a very
energetic earthquake could have occurred before the period covered by the available seismological catalogs. In this
context, the area needs more investigations to verify if an event with $M>6$ can repeat in the near future, as suggested
by the recurrence times of strong earthquakes generated by the typical Apennine extensional faults, specifically ~ 4
ky for this region (Galli 2020). Our study, points out the need of extending our detailed reconstruction of the buried
segments of the VCT, both along- and across-strike, and not only in the surveyed sector, but also on segments
bounding the northern edge up to the Mercure basin. In addition, our results may also be useful and preparatory for
orienting and locating further campains, integrating other geophysical tools, and possibly paleoseismological trenches,
drilling and sampling for dating (e.g., luminescence, radiocarbons, etc).
Such investigations could help to address the following crucial points, namely: i) ground truthing the presence of such
past coseismic surface breaks along the VCT, ii) define their timing, iii) obtain a recurrence time for the most energetic
events, and ultimately iv) aid a quantitative evaluation of the probability that a strong earthquake could hit the area in
the near future.
**6. Conclusions**
Our workflow allowed for the creation of a detailed 3D model which reconstructs the near-surface pattern of
paleoseismic ruptures across an area straddling the VCT active fault across the Campotenese continental basin (Mt.
Pollino region). Based on previous geological mapping, the VCT in southern section of the study area was
hypothesized to be buried under a clastic Holocene cover.
We have used non-destructive GPR survey, a powerful tool to investigate the shallow geological structures. The
processing, analysis, assemblage, and interpretation of the 49 GPR profiles was pursued using expertise, techniques,
and tools borrowed from reflection seismic industry applications. The use of GPR has allowed us to quickly investigate
the study area with low costs, in a non-destructive manner and without special authorizations. A relatively fast pseudo-
3D GPR survey operated during four days of team fieldwork was an efficient compromise between spatial coverage
of the study site and duration of the acquisition. On the other hands, the data processing was non-trivial, requiring
about six months overall to set up an optimized workflow, due to challenging characteristics like the steep and rugged



topography as well as sharp lateral variations of geophysical properties of media (Triassic Dolostones vs Quaternary
deposits).
More in detail of the study area, our structural reconstruction shows several sets of sub-vertical discontinuities (1-4 m
depth range), which we interpreted as extensional surface faulting, bounding small local "graben or semi-graben-like"
structures located within Holocene sediments and down to underlying dolostones. We have also identified some
chaotic and laterally discontinuous GPR-stratigraphic facies, interpreted as near-fault deposits (i.e. colluvial wedges
?). These structures suggest the possibility that surface faulting occurred in relatively recent times, but its traces were
successively leveled by the concurrent natural processes of erosion, aggradation and, anthropogenic activities. All
these buried features suggests that past strong earthquakes ($6 < M_w < 7$) might have occurred in the study area, which
is located within the central sector of the well known "Mt. Pollino seismic gap". As in the time range covered by the
historical seismological catalogs there is no record of such energetic events, we hypothesize that the area could be at
high risk of occurrence of a strong earthquake.  We hope the primary effect of our work is to raise the level of attention
regarding the seismic hazard in the Campotenese area, thus prompting further research to achieve an improvement of
the base-knowledge for assessing its seismogenic potential. We firmly promote a more widespread use of our GPR
workflow, particularly where near-surface data are scarce, as a base study for other seismic gaps worldwide.
**Acknowledgments**
The study has benefited from funding sources: Agreement INGV-DPC 2012-2013 & 2014-2015, Project S1 –
Miglioramento delle conoscenze per la definizione del potenziale sismogenetico - Base-knowledge improvement for
assessing the seismogenic potential of Italy, https://sites.google.com/site/ingvdpcprojects1/home, resp. Cristina
Pauselli, funded by Italian Presidenza del Consiglio dei Ministri – Dipartimento della Protezione Civile (DPC). The
paper does not necessarily represent DPC official opinion and policy. We sincerely thank Leonardo Speziali, Prof.
Costanzo Federico, and Roberto Volpe for their support during the field operations, as well as Khayal Gavarof for his
valuable collaboration in data processing. We thank QGIS (https://www.qgis.org/it/site/) for providing the software
with an open-source license, Petroleum Experts (https://www.petex.com/products/move-suite/), and dGB
(https://www.dgbes.com/) for providing the academic licenses MOVE and OpenDtect software. We acknowledge
NETGEO for academic access to the NRTK network (http://www.netgeo.it). We would also like to thank the
Ministero dell'Ambiente e della Tutela del Territorio e del Mare (MATTM) and the Regione Calabria for providing
free access to geospatial data such as DTM and aerials (Regione Calabria - www.regione.calabria.it, under license
IODL 2.0. - https://www.dati.gov.it/iodl/2.0/). The paper is the results of collaboration within the framework of the
Interuniversity Center for 3D Seismotectonics with territorial applications - CRUST (https://www.crust.unich.it/). The
GPR dataset presented in this study is available on request from the corresponding author.



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




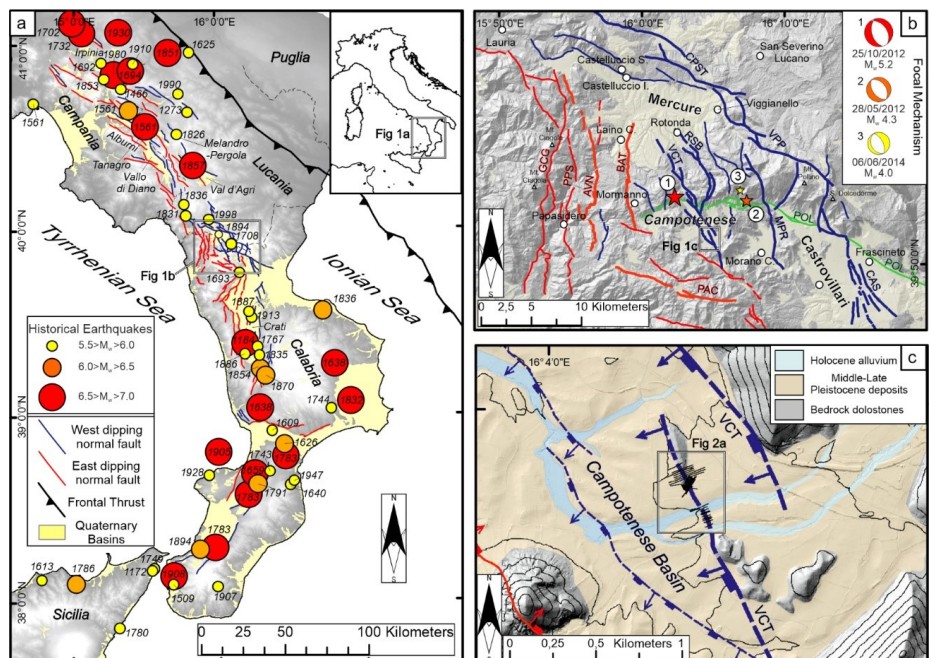

**Figure 1**


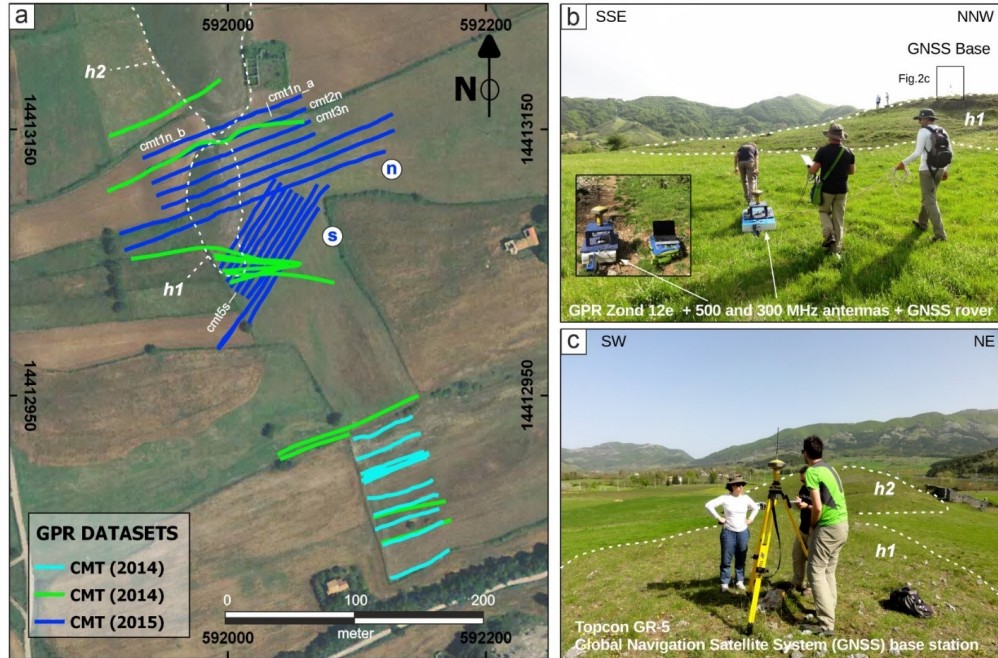

**Figure 2**



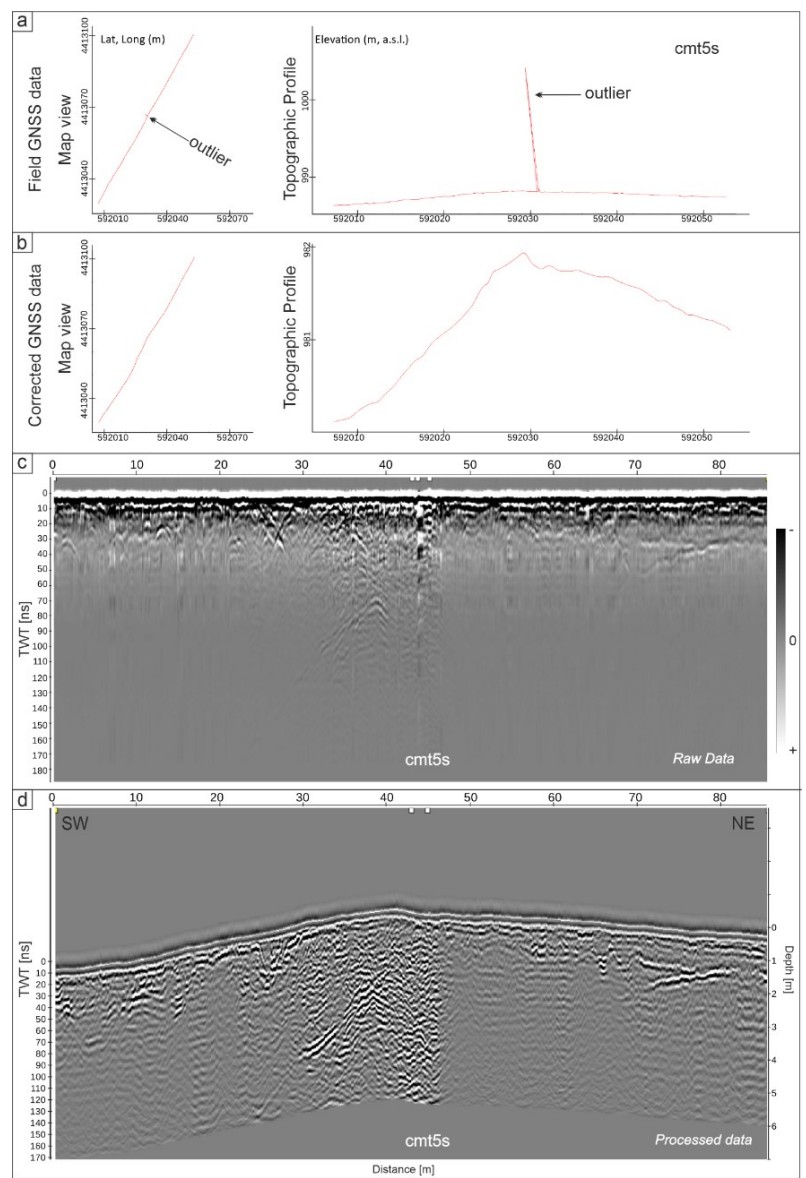

**Figure 3**


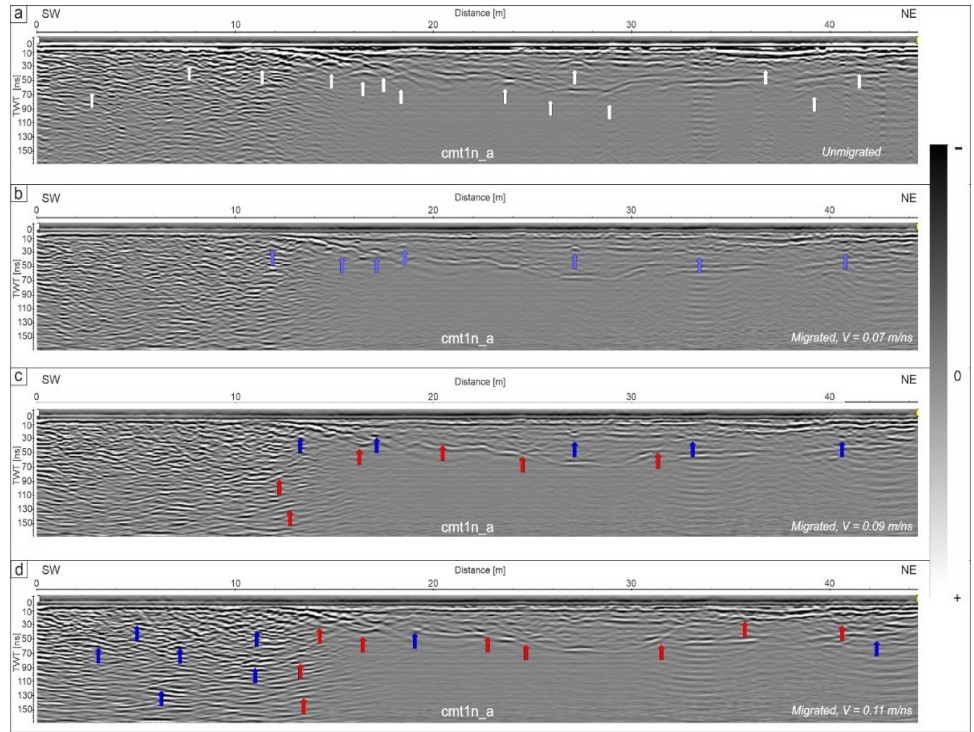

**Figure 4**

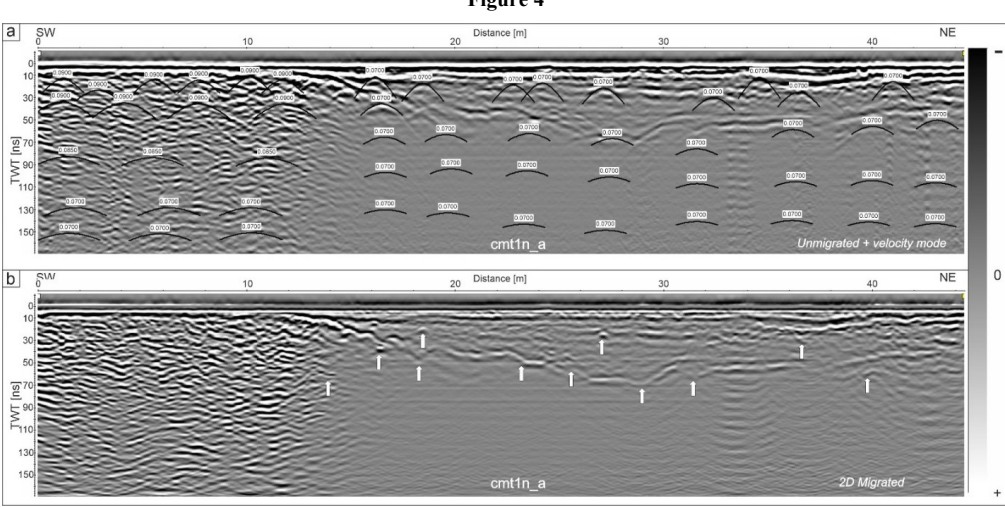

**Figure 5**





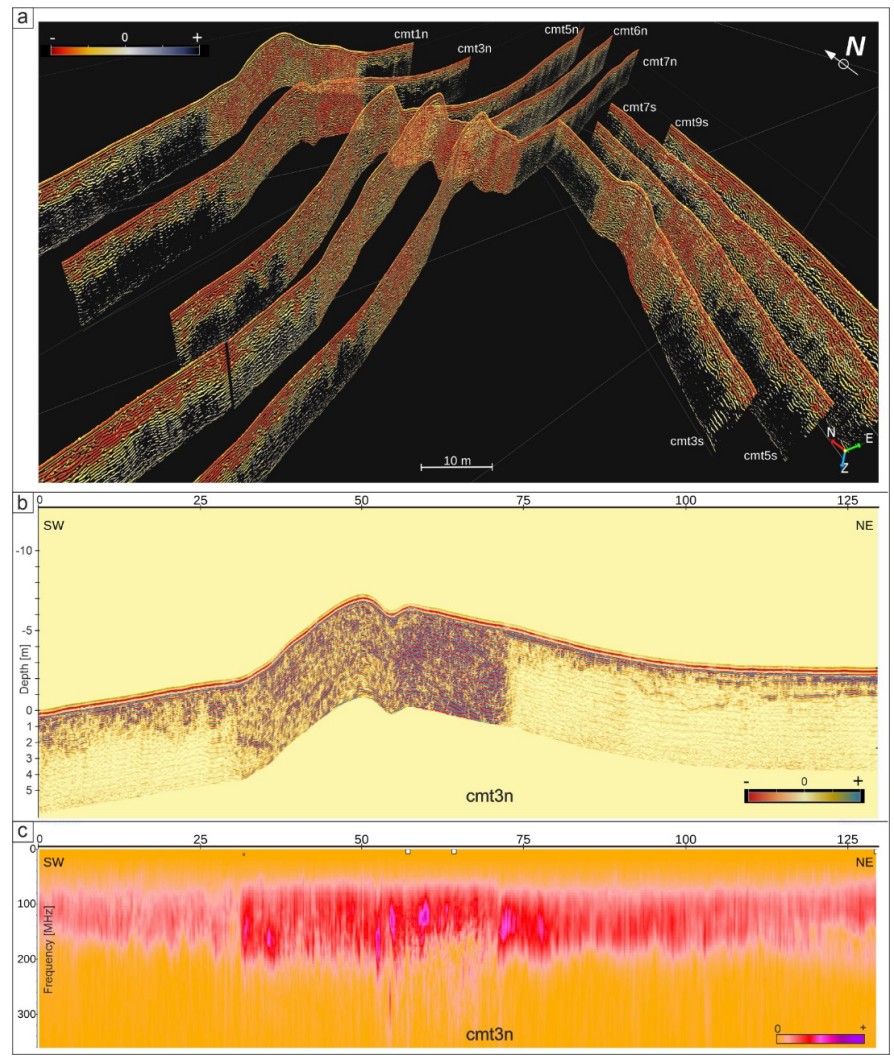

1009

**Figure 6**





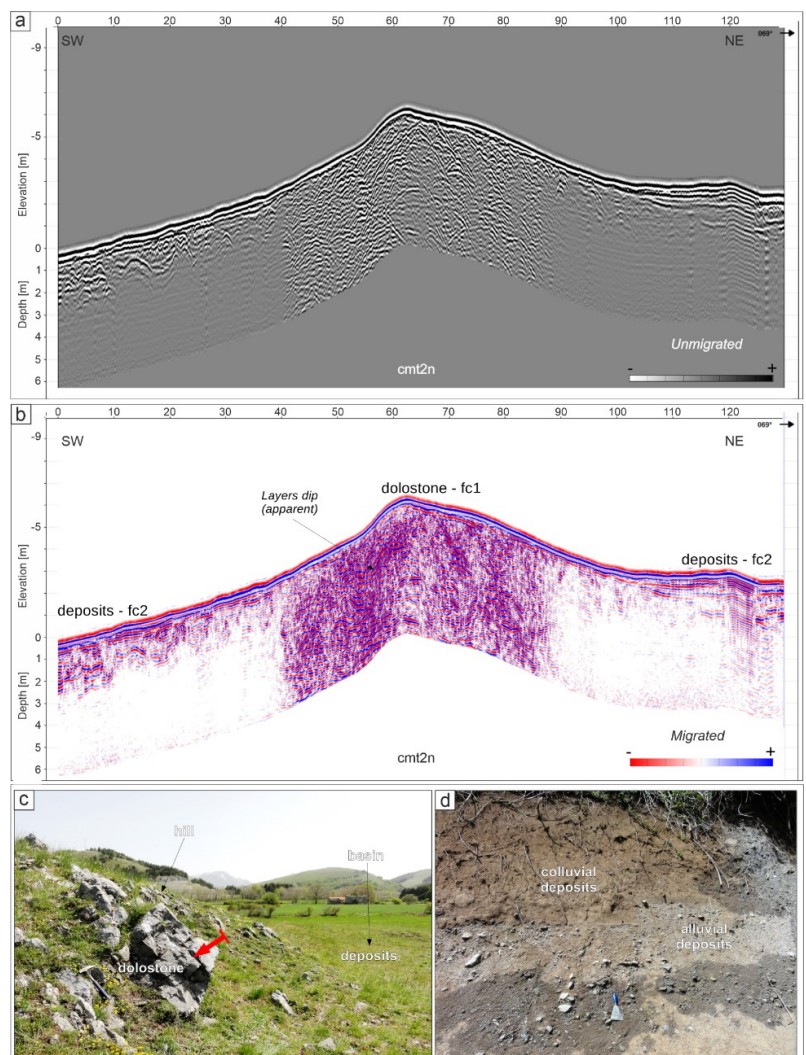

1010

1011                                                    **Figure 7**



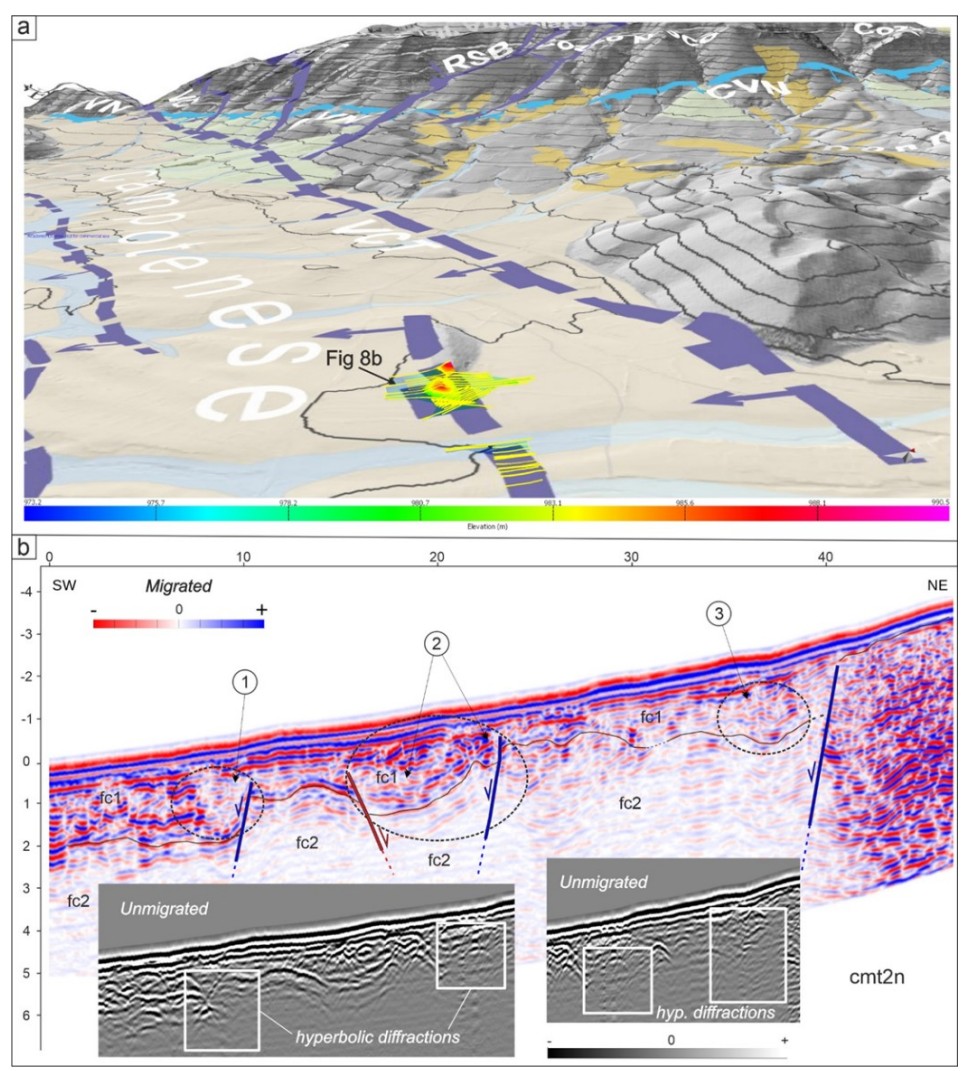

**Figure 8**





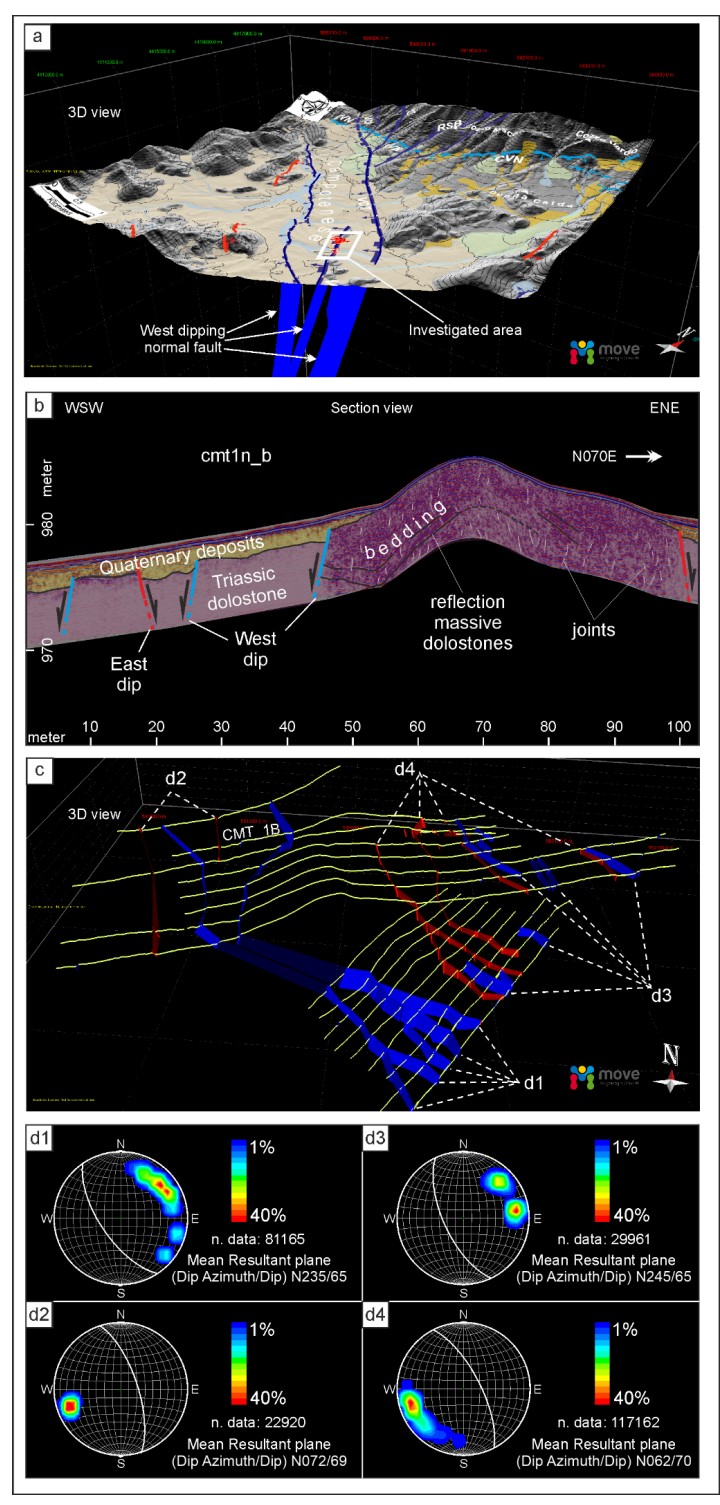

**Figure 9**




**Table 1**

| GPR survey information and parameters | | |
|---|---|---|
| Antenna frequency (MHz) | 300 (preferred) | 500 |
| Number of acquired profiles | 45 | 4 |
| Total profile length (m) | 3789.5/4153 | 363.5/4153 |
| Profile distance (m) | 10 and 25 (in g1 and g2) | not regular |
| Traces distance (m) | 0.05 | 0.02 |
| Number of samples | 1024 | 512 |
| Time window (ns) | 300-200* | 200-100* |


**Table 2**

| Processing Flow | Parameters (300 MHz) | Parameters (500 MHz) |
|---|---|---|
| Trace editing, coordinates editing and corrections | - | - |
| Time-zero correction | - | - |
| Dewow (ns) | 10 | 5 |
| Amplitude recovery function: $g(t)=(1+a*t)*e^{(b*t)}$ | linear: 0.5 (2014) & 1.2 (2015) exponent: 0.15 (2014) & 0.6 (2015) | linear: 0.5 (2014) & 1.2 (2015) exponent: 0.15 (2014) & 0.6 (2015) |
| Velocity analysis | Diffraction hyperbola fittying | Diffraction yperbola fitting |
| Background removal (ns) | Applied from 5 ns to end (computed on all the traces) | Applied from 5 ns to end (computed on all the traces) |
| Bandpass filter (MHz) | 32/96/650/700 | 64/112/750/800 |
| F-K filter | customized | customized |
| Time migration (2D Kirchhoff) | 2D velocity models | 2D velocity models |
| Topographic correction | GNSS/GIS Elevations | GNSS/GIS Elevations |
| Time-depth conversion (Quaternary deposits) | v = 0,7 m/ns | v = 0,7 m/ns |


**Figures and Tables captions:**
**Figure 1 - Location maps of the study site (DTM sources: TINITALY by Tarquini et al., 2012 and by Regione**
**Calabria - www.regione.calabria.it, under license IODL 2.0. - https://www.dati.gov.it/iodl/2.0/): a) the image**
**illustrates the southern Italian peninsula with the regional faults pattern and the historical strong earthquakes;**
**b) map showing the studied region with local faults, and the detailed location of the three historical seismic**
**events (stars); c) detailed location of the GPR survey area within the Campotenese Quaternary basin crossing**
**a possible VCT fault splay.**



**Figure 2 - GPR acquisition campaigns: a) GPR grids collected at the study site during the three field visits**
**(aerial image source: Regione Calabria - www.regione.calabria.it, under license IODL 2.0. -**
**https://www.dati.gov.it/iodl/2.0/); b) acquisition phase using the 300 and 500 MHz antennae (in the insert) and**
**GNSS receiver used for accurate data positioning; c) GNSS base station set up during the fieldwork.**
**Figure 3: Topographic correction of GPR profiles: a) example of accuracy degradation of GNSS data,**
**displaying an outlier both in map view and in topographic profile, on which the positioning error is**
**considerable; b) GNSS coordinates and topographic profile after the correction; c) raw GPR section displaying**
**high reflectivity in the central sector; d) example of full processed profile with topography displaying various**
**reflection patterns encompassing dipping reflections and diffractions. Vertical exaggeration is 4.**
**Figure 4: Migration tests performed during the GPR data processing: a) unmigrated 2D GPR profile, 300 MHz**
**antennae; b) migrated profile using a constant velocity v = 0.07 m/ns, light-blue arrows indicate good**
**diffractions collapse; c) migration output obtained with a constant velocity v = 0.09 m/ns, with dark-blue arrows**
**suggesting good migration results (migration artefacts are shown by red arrows); d) migration results using a**
**constant velocity v = 0.11 m/ns, with dark-blue arrows highlighting good hyperbolas collapse, particularly**
**within the high reflective unit; red arrows highlight clear migration smiles.**
**Figure 5: Example of 2D time-migration of radar profiles: a) example of hyperbolic diffractions fitting used for**
**2D velocity model building; a constant velocity value has been assumed in deeper no-diffraction areas for**
**interpolation purposes; b) 2D time-migration results, highlighting the good performance of the process, which**
**collapsed the hyperbolic diffractions and restored reliable reflection geometry.**
**Figure 6: GPR data visualization: a) fence diagram showing the three-dimensional location of some**
**representative GPR profiles in the northern sector of the study site; b) bidimensional GPR profile displaying**
**the central high reflective sector and dipping reflections across the hill; c) spatial variation of a 2D amplitude-**
**frequency spectrum linked to variable physical properties of media along the profile. Vertical exaggeration is**
**4.**
**Figure 7: Correlation between GPR profiles and outcropping geology at the study site: a) unmigrated 300 MHz**
**profile displaying numerous hyperbolic diffractions; b) migrated profile displaying the apparent vs true layer**
**dip associated to fractured dolostone formation (facies fc1) and Quaternary deposits in attenuated sectors (GPR**
**facies fc2); c) Triassic Dolostone formation outcropping on the hill (red arrow indicates the bedding) and**
**nearby Quaternary deposits of the basin; d) an example of Quaternary colluvial and alluvial deposits**
**outcropping nearby the survey site. Vertical exaggeration is 2.5.**
**Figure 8: GPR data interpretation: a) three-dimensional image of the surveyed area, displaying the Dolostone**
**outcrops bounding the basin and the surveyed hill. The coloured surface represents the interpolated horizon**
**reconstructed after the picking of the interpreted Dolostone top reflection (DTM source: Regione Calabria -**
**www.regione.calabria.it, under license IODL 2.0. - https://www.dati.gov.it/iodl/2.0/); b) migrated radar profile**
**with the main interpreted fault splays and related sedimentary structures within the Quaternary deposits**
**(detail of unmigrated data in the two grey inserts): 1) semi-continuous and sub-horizontal reflections (fc1,**
**Quaternary deposits) onlapping the lower boundary (Dolostone top, black line above fc2); reflections package**
**thickening and truncation with localized attenuation, which suggest the presence of a colluvial-wedge close to**
**a W-dip normal fault (~0.6 m vertical offset); 2) mode discontinuous, from subparallel to wavy reflections**
**package (fc1) downlapping the lower Dolostone top boundary; the asymmetric, truncated reflectors thickening**
**is bounded by two possible conjugate normal fault strands (~0.5 m offset) displacing both fc1 and fc2; 3)**
**contorted reflections package with limited continuity, displaying thickening, truncation and distributed**
**attenuation, suggesting colluvial wedge deposits lying above fc2 close to a fault zone with a W-dip normal fault**
**(~0.8 m vertical offset). Vertical exaggeration is 2.**
**Figure 9: Results of the three-dimensional analysis and interpretation performed on the entire GPR dataset:**
**a) 3D structural model inferred after the geological mapping at the scale of the basin (DTM sources: TINITALY**
**by Tarquini et al., 2012 and by Regione Calabria - www.regione.calabria.it, under license IODL 2.0. -**
**https://www.dati.gov.it/iodl/2.0/); b) GPR section view with interpretation including synthetic and antithetic**
**fault splays; c) detailed structural scratch of faults obtained by the analysis and correlation across the entire**
**GPR dataset d) synthetic stereo-net plots of the fault planes in c), reporting the mean Dip Azimuth / Dip angle**
**extracted for the identified four main sets of discontinuities, with a Dip Azimuth ranging between N 235-245°**
**and N 062-072° for the for the west-dipping and East-dipping normal faults, respectively. Vertical exaggeration**
**is 2.**
**Table 1: Main information and GPR parameters used during the data collection (* the time window was**
**adapted depending on the surveyed area).**
**Table 2: Customized flow and details of the parameters used during the processing of the GPR dataset.**