# Peer review of "GPR signature of Quaternary faulting: A study from the Mt. Pollino region, southern Apennines, Italy."

_Solid Earth, 2021_

## Referee Comment (RC2)

[referee-annotated manuscript omitted]

---

## Author Comment (AC1)

**Reply to comment on se-2021-75**

Anonymous Referee #1

Referee comment on "GPR signature of Quaternary faulting: a study from the Mt. Pollino region, southern Apennines, Italy" by Maurizio Ercoli et al., Solid Earth Discuss., https://doi.org/10.5194/se-2021-75-RC1, 2021
* * *
Ref#1: This is a relevant paper with substantial contribution to the understanding of Italian tectonics and seismicity. A bonus to this paper is the 3D tectonic analysis, it is always good if authors attempt a 3D model as it inspires readers to think of new conceptual tectonic and geomechanical models. The study rationale is good, background history is comprehensive and informative. The discussion and conclusions could have been sharper…

Dear Referee #1,

we really thank you for your kind revision and appreciation of our manuscript. It was one of our aims to push the reader thinking on the 3D model following our pseudo 3D GPR survey results. We're glad to hear that the manuscript structure is good, and of course, we can make an effort to sharpen more the discussion and conclusions during the revision process.

Ref#1: … attempting to do less conjecture and put more numbers on age, displacement, slip rates and earthquake recurrence etc.

Authors: obtaining more numbers, as the referee suggests, would be certainly desirable. However, we are aware of the limits of the GPR method, we would prefer maintaining a conservative approach due to the unavailability of ancillary data (detailed chronostratigraphic constraints). As we have remarked in our paper, it is a pilot study, aiming to highlight in a non-invasive way the occurrence of strong earthquakes which possibly produced surface ruptures (now buried). We aim to stimulate geoscientists and institutions to achieve further studies providing new insights and "numbers" for the area in the next future.

Ref#1: - How recent is the faulting? Is there any age control? How many meters displacement were observed and what slip-rates could eventually be derived? What is the estimated seismic recurrence and risk? I see much conjecture, but little numbers. Are such numbers available or obtainable?

Authors: our geophysical data are currently the only available for the site, and direct data from trenching or sampling (and thus a precise dating of the deposits) are currently not available. In our opinion, the faulting might be recent, Holocene in age, as some interpreted discontinuities which break up the GPR reflections are very close to the topographic surface. If the referees and editor argue that it's worth (and not too speculative), we can eventually provide a tentative estimate of the displacements interpreted on each fault (and cumulated) breaking the GPR reflections and, a preliminary slip-rate based using chronostratigraphic assumptions. But we remark these results would clearly need a direct validation to confirm the presence and magnitude of such features. On the other hand, we think the estimate of seismic recurrence periods and risk is out of the possibility of this work.

Ref#1: - Why was 3D GPR not attempted, it could have answered many of the questions above?

Authors: as we mentioned above, this pilot study was the first of this type done in the area. The Campotenese dataset was not unique collected during the GPR campaigns, but we surveyed at least other four sites. Thus, we had to find a trade-off between the time spent for the fieldworks and the density of the GPR lines to collect, also to cover a wider region instead of only a super-dense, but more limited, area. On the other hand, we surely agree that a full-3D GPR survey, upon detecting interesting areas on 2D profiles, would be the next "obliged" step possibly before trenching. It might also contribute to better address some of the above questions as well as efficiently and precisely drive the excavation.

Ref#1: - About the strong continuous wavy and undulating reflections with much attenuation as mentioned on page 10 and in Figure 5 and 7; could these also be the base of soil slips or landslides deposits coming from the surrounding foothills? It is not seldom that such landslides occur under heavy precipitation and glide over conductive clay or muscovite layers.

Authors: we absolutely agree about the presence of conductive strata enriched in clay which are responsible for the stronger attenuation. The basin hosts an alluvial valley, so such processes might be common under heavy precipitations, as suggested.

Ref#1: - On page 12, en-echelon faulting is suggested as a mechanism in the 3D fault model. However, I believe that en-echelon faulting occurs only in strike-slip or oblique-slip systems. Where is the strike-component in the VCT fault system? I thought it was only extensional/normal faulting?

Authors: En echelon faulting is often attributed to strike-slip tectonics, but both natural observations and analogue modelling experiments reveal that this pattern also occurs frequently in extensional and contractional tectonic environments (e.g., Twiss and Moore 2007; Ferril et al 2016; Clifton et al., 2000; Davis et al., 2005; Otsuki and Dilov, 2005; Giba et al., 2012). In other words, in the recent literature, the term "en-echelon" lost the former genetic meaning defined in strike-slip tectonics, becoming a geometrical term which only describes an array of structures that partially overlaps along strike. Thus, we use "en-echelon" in this latter sense.

Ref#1: - Make the GPR sections larger with more zoom. I believe the data is high quality but it is not well visible in some Figures.

Authors: The GPR lines are frequently very long, thus using more zoom will reduce their length and cutting some data which means looking at the GPR reflections in a more limited context. However, we'll make an effort to try improving the zoom of the profiles, or possibly providing such profiles in a bigger format within the supplementary material.

Ref#1: - Figure 5: so the final migrations were 2D time migrations with variable 2D velocity models?

Authors: yes, confirmed. We initially spent time testing different migration algorithms, also using simpler using 1D velocity models. But we finally opted to customize each GPR profile its own 2D model which guaranteed superior imaging results.

Ref#1: - Annotate Figure 7 outcrops photos with interfaces and structures

Authors: thanks for the suggestion, we'll update the figure by adding more annotations.

Ref#1: - Figure 9c: what are blue and red faults again?

Authors: the blue and red surfaces build up the reconstructed fault network (blue= W-to SW-dipping synthetic; red=E-to NE-dipping antithetic faults) derived by the interpretation of each GPR profile. We will update the caption.

---

## Author Comment (AC2)

**Reply to comment on se-2021-75**

Anonymous Referee #2

Referee comment on "GPR signature of Quaternary faulting: a study from the Mt. Pollino region, southern Apennines, Italy" by Maurizio Ercoli et al., Solid Earth Discuss., https://doi.org/10.5194/se-2021-75-RC2, 2021
* * *
Ref#2: In this manuscript the Authors describe and interpret the Ground Penetrating Radar profiles acquired in different campaigns along a splay of the Fosso della Valle - Campotenese fault (VCT) in the Pollino Range (Northern Calabria, Italy), well known as a seismic gap region. Among the major objectives are the finding of evidence in the subsurface for Quaternary faulting along this buried fault and also the characterization of peculiar GPR signature of faulting in order to build a powerful methodology for areas of similar characteristics. Lots of data are produced on GPR acquisition and I found very interesting results and interpretation for the subsurface imaging of a sector of the Fosso della Valle-Campotenese fault (VCT) at the local scale. Manuscript provides a very high-quality data, and the inferences derived from the 3D are very well constrained. However, in my opinion several issues affect the manuscript and need to be fixed.

Authors:

Dear Referee #2,

we really thank you for your positive and very accurate revision of our manuscript. We're glad to hear that you like the data and results, and we are sure your corrections and suggestions will improve our work after the end of the revision process.

Ref#2: In general, i) the Authors use large parts of text, even redundantly and not always clearly, to refer to some general concepts not linked to the acquired data and to the aim of the manuscript. In different sections, there is confusion between what was found in the previous geological studies of the fault and what is new. This criticism has to be addressed to make the work sound in terms of objectives to be reached with the GPR acquisitions, the work core. ii) Similar problem occurs for the section GPR data description and interpretation that is very long and I would clean it by sentences on generic technical references (possibly to be insert into the methodology chapter) and shed light to the interpretation of the data.

Authors: Such a request is provided also by referee 1 and by the editor. The paper core is to provide not only hints on the structural setting and paleoseismicity of the study area but also to define and discuss, supported to proper literature references, a methodological workflow that can be widespread across similar regions and situations. We'll make our best to shorten the text where necessary and as requested.

Ref#2: Moreover, iii) at least the fault portion surveyed by GPR is claimed to be a buried fault, but then there is no a clear presentation on the mapping of the fault pattern at the surface and on which evidence it is based.

Authors: We display a 3D representation of the reconstructed fault pattern in figure 9c, aiming to provide a complete reconstruction of the main interpreted synthetic (i.e. W- to SW- dipping) and antithetic (E- to NE- dipping) structures, obtained by interpolating the GPR evidence of figure 8b (extended to other lines) and remarked in figure 9b. However, also following the comment of #Rev1, we figure out that it is not very clear. We'll surely update it, or adding a basemap to figure 9c or possibly providing an updated figure 9c. In the latter case, the figure will include the fault pattern, obtained by intersecting the subsurface fault model with the topography, displayed as a detailed structural map of the surveyed area.

Ref#2: This is critical also iv) to fully discuss the seismic hazard implication of the VCT fault based on the GPR results, as offset estimation, fault zone width, etc…rather than citing already well-established statements. Here, I expect you to explore the potentiality of the data relative to hazard assessment on the fault, including the definition of limits of the GPR approach on the recognition and dating of discrete events of faulting. This is why GPR signals remain very important for the preparatory phase of further investigations.

Authors: we agree with this consideration, and we think that, as we replied to Referee#1, based on the interpreted GPR profiles we can provide a reasonable estimation of GPR displacements, as well as an estimation of average fault zone width. The only possibility of going further, in evaluating the seismic hazard of the area, would require attributing the highlighted offset to a single coseismic displacement event. Evidently in the absence of direct subsurface data, this assumption would be too uncertain based only by using our data. As suggested, we agree it is an important point to specifically discuss the potentiality and limits of the GPR approach, particularly if without chronostratigraphic constraints and ground-truthing, as presented in this study.

Ref#2: v) Figure and figure captions need revision in order to become self-readable images.

Authors: We will revise the figure and figure captions also using your detailed comments

Ref#2: For the above criticisms in my opinion the manuscript is suitable for publication in SolidEarth with major revision. You may find several detailed comments, questions and proposed changes in the annotated attached file.

Authors: Again, many thanks for your effort, such detailed corrections and comments will considerably aid us during the revision process. We will provide a point-to-point reply to detailed comments together with the revised manuscript.

---

## Author Comment (AC3)

**Reply to comment on se-2021-75**

Luca De Siena (Editor)
Editor comment on "GPR signature of Quaternary faulting: a study from the Mt. Pollino region, southern Apennines, Italy" by Maurizio Ercoli et al., Solid Earth Discuss., https://doi.org/10.5194/se-2021-75-EC1, 2021
* * *
Dear authors

Both reviewers have positive comments, considering your study as impactful, with interesting results and interpretations. At the same time, there are some relevant questions you need to address in the revision. Both reviewers provide extensive information on how to amend the text and are clear about what they doubt. Please have particular care in tackling point I) of reviewer 2 - the text needs shortening. I will provide a final revision of the revised paper after the reviewers, focusing on this issue.

Dear editor,

we really thank both reviewers for their positive and constructive comments, which surely will improve our final manuscript. We'll take care of their suggestions in the revised manuscript. Thank you also for your kind support, we'll make our best to shorten more the discussion and conclusions to aid your final revision.

All the best,

Maurizio Ercoli and co-authors

---

## Author Response (AR1)

**Point-to-point to comments on se-2021-75**

Editor and referees' comments on "GPR signature of Quaternary faulting: a study from the Mt. Pollino region, southern Apennines, Italy" by Maurizio Ercoli et al., Solid Earth Discuss., https://doi.org/10.5194/se-2021-75-RC2, 2021
* * *
**#editor and #referee1**

Dear all,

as we've already replied during the discussion phase, we really thank you for your kind revision and suggestions. Some replies to your general comments were already addressed within the discussion, we have answers to more specific requests in the following document, as many points overlap with ones of #referee2. During the revision process, we have made an effort to improve and sharpen the introduction, interpretation, discussion and conclusions, providing more quantitative information but also clearly defining the limits of our workflow. You'll find all the answers here and the tracked corrections in the marked manuscript. We trust that this revision will make our paper a strong candidate for publication in Solid Earth.

**#referee2**

Dear #referee2,

Again, thank you for your detailed review and associated suggestions. We've made a considerable effort to address all your requests and suggestions and we think the manuscript has benefited with the changes. We hope this will be enough to consider the paper for publication in SE.
* * *
Ref#2 lines 13-15: evidence of late Quaternary faulting across the fault not across the basin. Please present the fault clearly... name, location and portion investigated with GPR. any fault in the area? see previous comment

Authors: we agree with the comments, we added such information in the text.

Ref#2 line 27: what do you mean for near-surface data: you have surface data but scarce is the knowledge of the subsurface? you need to be clear in any single words of the abstract

Authors: yes, correct, with near-surface we meant the portion of shallow subsurface (tens of meters) typically investigated with high-resolution methods (like the GPR).

Ref#2 lines 32-34: please revise the definition for seismic gap. It is not correct. To be added that the gap region is surrounded by regions struck by large earthquakes in historical or recent times....the presence of faults and of their seismic past activity, of their possible quiescence as well, is from morphological and paleoseismic data.... also gaps are candidate regions for the occurrence of large earthquakes in the near future ||| For the seismic gap definition you need to cite McCann et al., 1979. Galadini and Galli 2003 applied the theory in central apennines, but before in time Cinti et al 1997 recognize a seismic gap, silent area in the castrovillari zone....

Authors: thanks for the comments, very appreciated. We have changed the text following the advice and updated the citations.

Ref#2 line 37: provide some examples

Authors: citations were provided above so we have shortened the introduction.

Ref#2 lines 39: not clear. please rephrase

Authors: done.

Ref#2 lines 39-40 and references comments untill line 47: As a general comment: too much info on this sequence, and it is not needed to confirm that paleoseismology is useful, and moreover paleoseismology is not the topic of this paper. Suggestion is to focus on the precious info from GPR on fault and also that it is complementary to trenching. And particularly the added value this technique provides to better understand the active faults in the Pollino range area.

Authors: we have significantly reorganized and shortened the text, updated the text to the most representative literature, aiming to highlight the central Italy sequence might be an analogue seismic gap for the southern Italy.

Ref#2 lines 54-59: you are referring to trenching. rephrase || unclear what you mean.

Authors: we have rewritten the text referring to trenching.

Ref#2 line 61: this is a quite dated paleoearthquakes inventory of Italy, is this you want to cite?

Authors: we have removed the citations as requested.

Ref#2 line 63: this is a report for Civil Protection Dept.

Authors: yes, we cite both reports referring to the project results and provide the URL in the reference list.

Ref#2 line 68: and what about southward?

Authors: we have rewritten the text.

Ref#2 line 70: there is confusion: Mercure and Campotenese basin are bounded by faults, the Castrovillari plain (I would not refer to it as a basin) is bounded by the Pollino fault and crossed by the Castrovillari fault systems. There is the signature of the faults in any case. Please attention to the description of faults setting in the gap area and to the morphology as well.

Authors: we have rewritten the text.

Ref#2 line 77: this is a submitted paper!

Authors: we have updated the citation; the pre-print is online and in revision in SE (https://doi.org/10.5194/se-2021-76) within the same special issue. URL: https://se.copernicus.org/preprints/se-2021-76/

Ref#2 line 80: If I understood well this is a buried or better hidden fault. So, how was recognized in the published works, parameterized, length at surface assessed......This is crucial! I understand erosion and cover but if the fault does not have surface expression anywhere along 15 km and you see offsets at the subsurface, it could likely mean the amount of offsets are not large enough and also very long recurrence avoiding the cumulation and preservation at the surface.

Authors: the parametrization of this fault is already documented in Brozzetti et al. 2017, but we have briefly integrated the results into the text. The authors remark that this fault is well visible within the Mercure basin as a 3 km long and continuous scarp carved on the present topography. They also suppose its continuation towards South along the VCT being buried below the Holocene alluvial deposits and occasionally suggested by geomorphic scarps. Such observations provided are one of the main motivations for GPR surveying. A dedicated sentence is present also within the Tectonic setting and seismicity in the submitted manuscript: "Along the E-side of the Campotenese basin, the VCT is generally buried by Holocene deposits, but its localization can be inferred based on stratigraphic observations and geomorphic features, such as sharp ridge fronts, linear scarps, and slope breaks".

Ref#2 line 81:

Authors: "Fosso della Valle" has been added in figure 1c.

Ref#2 line 83: You just cite these papers few words before....really over referencing (particularly of papers by the authors of this manuscript).

Authors: text rewritten.

Ref#2 lines 83-84: please explain which is the long-term surface expression of the VCT, which are the evidence for current activity. These are data at the basis of your paper.

Authors: we have integrated and provided more details in the text within the chapter two (Tectonic setting and seismicity).

Ref#2 lines 84-85: I suggest to add the 1915 Fucino clear case of surface faulting in the Apennines, testiified by: Oddone, E., 1915. Gli elementi fisici del grande terremoto marsicano-fucense del 13 gennaio 1915. Boll. Soc. Sismol. Ital. 19, 71–216. || these citations refer to the same earthquake (Norcia). Insert only one of these. In general, avoid too long list of citations when not needed and relevant.

Authors: text changed following the suggestions.

Ref#2 line 92: ??to rephrase

Authors: text changed and simplified.

Ref#2 lines 95-98: I do not understand. why to say this and here. The process described is something that always happen! The critical point to raise concerning the preservation of surface faulting expression in the stratigraphy is the rate of occurrence of surface faulting, the size of the displacement vs the rate of deposition and/or erosion.

Authors: We agree with the suggestion, so we have removed the sentence from this section.

Ref#2 lines 98-99: I think you may summarize the objectives. Some are just consequential.

Authors: We have reorganized and modified the text. We maintained the difference between i) and ii) to distinguish a GPR methodological workflow vs the specific application providing new insights on the VCT fault.

Ref#2 line 117: just in german language?

Authors: yes, and we have updated the citation.

Ref#2 line 131: maybe you mean historical seismicity and ....because it is very hard with a seismological study to associate past earthquakes to a specific fault.

Authors: we replaced "several seismological" with "recent seismic activity".

Ref#2 lines 133-134: again, the following citations are not all strictly appropriate for the relative affirmation. Here you clearly mention to Quaternary basin and to seismological and paleoseismological investigations but not all the papers you list are relevant for this. Moreover some papers focus on the same area and also redundantly and useless cited all over the text. || Galli et al. 2006 ?? not in the reference list

Authors: we have reduced the citations.

Ref#2 line 137: Here you cannot ignore the compressive regime with an east–west to east-northeast/west-southwest maximum horizontal stress in the Ionian offshore. The Sibari coastal area within the Pollino seismic gap region represents a transition zone where also compatible oblique normal-lateral faults have been recognized in the literature (i.e., the Sybaris fault in Cinti et al. 2015 and the Rossano fault in the surrounding area in Galadini et al., 2001).

Authors: we have added a sentence with a reference to the suggested papers.

Ref#2 line 138: Warning: you investigated a fault splay within the area. This is not the project where probably you were looking at a different scale with different approaches.

Authors: we have changed the text, as requested.

Ref#2 lines 143-144: Say something more about the Rotonda-Campotenese fault, is it a master fault and the VCT is a fault splay in its hangingwall. How far are they, the VCT plain joins the master fault at which depth? Your data in case are on a secondary fault and the faulting history (expression, recurrence etccc) needs to be discussed with this geometry in mind.

Not marked in Figure. Needed to be highlighted the Rotonda-Campotense fault and the VCT splay.

please revise the figure showing this pattern.

Authors: we have added a more complete description of the faults as well as improved the figure, as requested.

Ref#2 line 147: Reference needed

Authors: We deleted the sentence to shorten the section, as we evaluated it too specific for the tectonic setting.

Ref#2 lines 157-159: Who made this analysis? what about the 1693? see Tertulliani and Cucci 2014 on the similarity with the modern sequence...

Authors: We have removed the sentence to shorten the section and because the analysis was not in the focus of our work.

Ref#2 lines 160-163: But this was an objective of the work!!! anticipated.

Authors: we agree, we have removed these sentences.

Ref#2 lines 174-175 and 185: move the reference at the end of the sentence or not have sense to be used. || these are both papers on gpr applied to archaeology, cannot be used just one?

Authors: corrected

Ref#2 lines 190-195 for example, this paper is relevant to cite in this context?

Authors: Yes, the two highlighted citations are among the most recent and interesting literature examples of GPR application on faults.

Ref#2 lines 199-201: e.g., this is not a complete list of papers for this reference, then I would cite only the most representative, or the most recent, or the oldest.... || more than one of these papers better refer to the sentence below (line 202) || what about leaving 2010 and reference therein? check other parts in the text where you repeat these papers.

Authors: we have removed the proceedings, the others by Gross and by McClymont are in our opinion all fundamental papers to cite for GPR applications on faults.

Ref#2 line 214: i.e. loose and chaotic material? colluvial wedges not necessarily provides the same conductivity. For example, if buried and old could be cemented....

Authors: we agree with the observation, we have removed consideration in the methodological chapter.

Ref#2 line 215: what do you mean for near-surface faulting?

Authors: we meant the subsurface sector covered for example by GPR survey; it is referred to near-surface geophysics (for example, a definition is provided by Dwain K. Butler, January 2005, What Is Near-Surface Geophysics? Near-Surface Geophysics, 2005, 1-6, https://library.seg.org/doi/10.1190/1.9781560801719.ch1). We have modified the text with "near surface interpretation of faulting".

Ref#2 line: explain where did you acquire relative to the VCT traces in Fig.: along the western splay of the fault and covering a fault sector of X km along strike and a max of 200 m across?

Authors: we have rephrased the sentence including more details as requested.

Ref#2 lines 238-239

Authors: corrected

Ref#2 lines 245-247

Authors: we have changed the text, providing a better explanation of this concept.

Ref#2 line 250: Results? you are still in the survey section.

Authors: we referred to a preliminary interpretation which was useful to better plan the last acquisition. However, corrected.

Ref#2 line 257: which grids?

Authors: fixed

Ref#2 line 267: why favourable? elevation and no forested area?

Authors: corrected

Ref#2 line 280: 3a and 3b not cited yet.

Authors: now cited, and we've upgraded the citations.

Ref#2 lines 383-384: so why you claim wavy reflection as a key layer for this research, explain in which sense or rephrase.

Authors: we have edited and simplified the sentence to avoid repetitions.

Ref#2 line 390: move at the end of the sentence. Is it what you observed in the case of Mars, right?

Authors: corrected

Ref#2 lines 393-398: not clear is a suggestion to consider?

Authors: we corrected the entire paragraph as requested

Ref#2 line 422: see comment on figure caption. The estimates of the offsets by the radargrams are critical and main subject of the paper. Please, show how you proceed for measuring (which surfaces were used to project across the fault) and also explain if you observe an increasing of the amount with depth.....although subtle evidence but this would imply multiple events. to discuss!

Authors: at this stage of the base-knowledge for the study site, our main goal was to find and highlight peculiar GPR signatures in the Quaternary deposits to possibly interpret as faulting features. Nevertheless, we agree that an additional quantitative estimation of the fault displacements would be the best result to achieve, but we are aware it is a critical point particularly without a direct field validation. For this reason, we would avoid a very detailed parametrization deriving by possible overinterpretations of the GPR features. Keeping in mind this consideration but following also the #ref2 (and #ref1) comments, we provide a tentative estimation of the interpreted GPR displacement for the representative profile cmt2n. Thus, we also have integrated in the text and figure 8 some details about the procedure we used, as requested. Regarding the last request, it is currently not possible to know whether the associated displacements occurred during a single surface faulting event or during multiple events.

Ref#2 line 443: higher respect to which other resolution? you mean the comparison between 9a and 9c? well, more than a resolution issue is a different scale of observation.

Authors: we have corrected the text, detailing that the scale of observation is metric.

Ref#2 lines 444-446: ? maybe figure 9c to cite || (at a scale of....) Specify what you want to remark. A detailed mapping of the fault lines with focussed GPR acquisition particularly for such buried sector of the fault?

Authors: corrected and integrated, then moved to section 5.1, in which we detail the description of the 3D model using figure 9.

Ref#2 line 452: associated? explain

Authors: we meant faults-splays of the VCT within the Quaternary sediments. However, we have almost completely reorganized and rewritten the chapter 5.1, in order to remove redundancies and shorten the text as requested also by the editor.

Ref#2 line 456: no name to refer?

Authors: we have referred to h1 an h2 labels we introduced for the hills in figure 2a and related caption.

Ref#2 line 457: fault name?

Authors: we have rewritten the text.

Ref#2 lines 475-476: to be rephrased

Authors: we have rephrased the sentence

Ref#2 lines 479-481: try to rephrase

Authors:   we have rephrased the sentence

Ref#2 lines 491-492: please mention, associated based on what.

Authors: we have rephrased the sentence

Ref#2 line 502: more than that to the north and about 50 m to the south. For clarity, you may mention the year of the earthquakes....

Authors: text revised and years added.

Ref#2 lines 511-515: very twisted. rephrase

Authors: we have rephrased and refined the paragraph reorganizing it and removing some redundancies.

Ref#2 lines 516-519: you jump too much without clarity on the concept. A problem is that here you are writing a seismic hazard implications chapter using all affirmation that are independent from the results obtained in this paper.

Authors: we have modified the sentence, which clearly introduces in the discussion other recent works supporting this idea but which are in agreement with our GPR results.

Ref#2 line 519: I suggest to mention about geodetic data and refer to the different papers providing interpretation of tectonic strain and earthquake recurrence in the Apennines. Completely missing in your text on seismic gap.

Authors: as we do not basically provide any info on the strain or recurrence data in our work, we have removed the sentence.

Ref#2 lines 519-523: all this should go in different chapters, not here in hazard. In part in 6. conclusion. and the utility of the gpr data as preparatory phase for other type of campaigns should go at the beginning of the paper as a motivation of the application.

Authors: we removed the sentences as suggested, however we referred not only to GPR results but to a series of additional analysis that will be useful as a further step to confirm our results and obtain quantitative information. We report this concept in the introduction as suggested and in the conclusions as well.

Ref#2 lines 549-551: Literature is full of papers that claim this!! It is an implicit concept of the seismic gap definition. change in: Our study confirms the presence of seismic potential and thus the possible occurrence of a large etq in a future. Remind that: with your workflow you do not extract any date for events and no succession of past events....

Authors: We agree with this comment. We have added in the text the need of further studies to define timing and recurrence time (not provided by GPR).

Ref#2 line 551: please no risk but in case hazard...

Authors: We agree, the difference is clear, it was only an error as we meant "high probability of occurrence". We have corrected and shortened sentence.

Ref#2 line 572: to be carefully revised according to the changes suggested in the text.

Authors: the list has been revised according to comments.

Ref#2 line 1025: -highlight all the splay of the Rotonda-Campotenese fault that includes the VCT splay. -not visible the rectangle for 1c area, try with a different color line - focal mechanisms from Pondrelli et al 2002 and Scognamiglio et al. 2006? || In figure c, as well as in figure b, you have different lines for faults with different symbology. Moreover what VCT stay for. GPR traces are barely visible, change color and enlarge or useless.

Images should be self support by the caption, without reading the text.

Authors: corrected according to the comments: RSB and VCT are now enhanced in fig.1b; the rectangle of the area is now much better visible; we have updated the legend of fig. 1c, to explain the different symbology used, also to highlight the main and minor faults; the GPR lines have been removed from this figure; now the VCT fault name is reported also in its extended version. The caption has been updated.

Ref#2 line 1028: cmt stay for.....

Authors: added in figure caption.

Ref#2 line 1034: to be insert in the text. not cited.

Authors: added.

Ref#2 line 1037: white arrows?

Authors: added.

Ref#2 line 1045: white arrows point to....

Authors: added.

Ref#2 line 1050: please insert this info also within the image

Authors: added in all images displaying the depth on the y axis.

Ref#2 lines 1055-1057: view to the SW? || the arrows is not the best to show bedding. Try something different. || please insert this info also within the image

Authors: image corrected by changing symbols and adding labels as well as two informative inserts. Vertical exaggeration added in figures a and b. Caption corrected accordingly.

Ref#2 line 1061: I like it a lot! Please, add the horizontal/vertical scale meters.

Authors: Thank you! corrected.

Ref#2 line 1065- 1066: do you see a wedge shape from the signal? in case it is nice to mark it with a symbol. But I am wondering if you mean a deposits from earthquake fault free-face degradation in the close HW of the fault, that felt and stucked within a small graben coseismic area (that I see in the section shown area 1, don't you?)

Authors: We agree about another possible interpretation of area 1. Our aim is only to suggest within the dashed circles, avoiding overinterpretations, a few areas with interesting geometry possibly resembling "colluvial wedges" or possibly, as suggested, deposits from earthquake fault free-face HW degradation. Thus, we have added a label "cw?" to suggest the presence of such geometries. Vertical exaggeration label was added as well.

Ref#2 line 1066 show how measured. interpolation of surface across fault...

Authors: we have updated the figure adding the inset to show how we estimated possible GPR displacement of GPR reflections and integrated the text. We hope this solution can work fine to clarify this process.

Ref#2 line 1068: the 0.5 is for fc2 and fc1? no larger offset for fc2? show how measured. interpolation of surface across fault.. it is a net measure? can you estimate an offset for fc1?

Authors: we partly already replied in the above comment, adding the inset and integrating the text and figure caption.

Ref#2 line 1068: no fault line to trace?

Authors: we picked the potential faults which following the geophysical indexes mentioned in the text are more visible, improving our interpretation by updating the figure.

Ref#2 line 1070: mark the cw with a symbol

Authors: "cw?" labels added in figure

Ref#2 line 1071: show how measured. interpolation of surface across fault.. || please insert this info also within the image

Authors: done as answered above.

Ref#2 line 1073: toponyms are not clearly readable. Again, abbreviations legend is missing: CVN. RSB, VCT......   The light blue line?

Authors: the abbreviations of faults surrounding the study area are defined in the text and reported in figure one. The light blue line is now mentioned within the figure caption and updated in all figures.

Ref#2 line 1073: This is a 3D view at the scale of the basin with geology and faults from a different source paper, not this one, please cite. Figure 9a could be imaged without the GPR results from this work, that cover only a portion of the fault and do not provide info for the model at this scale. I would not present this illustration as a result.

Authors: We agree the image is modified using a map after Brozzetti et al. (2017a) but here it is presented including a 3D model of the main fault segments. We think it is necessary to aid the readers better figuring-out with the different scales of observation from the basin to the very detailed GPR scale.

Ref#2 line 1075: nice! I do not understand why do not show the same interpreted view for cmt2_n in figure 8 (as a fig. 8c). Structures identified are the same and also deposits. Here you insert a new section without any apparent reason. please explain. However here I would leave the 3D.

Authors: thank you. We here present the GPR results using another profile (figure 9b) to show more data and the similarity/coherency of interpreted features, to convince the readers about the spatial continuity of the structures detected. Finally, we summarize the interpretation as a detailed 3D model in figure 9c (we agree to leave this image), but we also enriched the figure with a conventional structural map with more quantitative information. We thrust the updated image fitting all the requests.

Ref#2 line 1075: the blue fault around meter 25 on cmt1_bn is alone?

Authors: yes, we interpreted such fault as clear only on this profile, and we evaluated not enough the faulting GPR signature on adjacent profiles to interpolate a longer surface. To consider, however, that the GPR profiles are to the north (n) are 25 meters spaced, whilst 10m on the south (s), thus allowing more detailed insights.

Ref#2 line 1076: Scale of observation? How large and long is the area you show. it is needed. Also, it is to avoid confusion with the model in fig. 9a. Symbology to be explained: d1 to d3? Figures have to be self-readable.

Authors: we agree with the comment. As reported in the comment above, we wanted to highlight the different scales between figures 9a-9c. Figure corrected.

Ref#2 line 1080: please insert this info also within the image

Authors: together with other labels, also the "vertical exaggeration" was added in figure.

---

## Author Response (AR2)

**Point-to-point to comments on se-2021-75**

Reply to Referee #2 and editors on "GPR signature of Quaternary faulting: a study from the Mt. Pollino region, southern Apennines, Italy" by Maurizio Ercoli et al., Solid Earth Discuss., https://doi.org/10.5194/se-2021-75, 2021
* * *
**To: referee #2 and editors**

Dear all, we really thank you for your final revision and for accepting our manuscript for publication. We here provide our answers to you requests.

Thank you,

Maurizio Ercoli, Daniele Cirillo and co-authors.

**To: referee2**

In my opinion the revised version of the manuscript is substantially improved in text and figures and ready to be accepted for publication. Below few minor additional suggestions/comments.

Dear #referee2,

Again, thank you for your positive comments. We are glad that the changes applied to our manuscript now fits with your requests and suggestions.

We have applied also some minor corrections, as requested.

Best regards,

Maurizio Ercoli, Daniele Cirillo and co-authors.
* * *
REV#2:

Page 12 line 37: check the sentence sound, I suggest to change in: "But in cases like the VCT, the GPR data assume a key-value since provided key fault parameters where no direct information on the nature of the surveyed deposits and no accurate dating is available"

Authors: corrected

REV#2:

Page 13 line 1: I do not think GPR technique will never directly extract date for single event: dating of an earthquake (absolute but even relative) requires sampling of the stratigraphy at different levels.

Authors: we agree, thus we removed "currently" and we have rewritten the sentences 35-36 of page 12.

**To: referee #2 and editors**

TE:

I have a couple of final remarks, which I would like you to take into account:

Lines 22-23: The sentence is hard to read, possibly split - "Pseudo-3D GPR data are harder to interpret, as they require to link geophysical features among different radar images. However, their use enabled the reconstruction of a reliable..."

Line 24: Rewrite more simply - "Our contribution better characterizes active faults in an area which falls within the Pollino seismic gap and is considered prone to severe surface faulting."

Line 26: The sentence remains unclear, try: "Our results are a constraint for further research at the study site while our approach and workflow are ideal for similar regions characterized by high seismic hazard and scarcity of near-surface geophysical data."

Authors: we have modified the sentences in the abstracts following the comments and rewriting the text.

Authors: Dear editors, finally, we would like to add also Dr. Daniele Cirillo (second contact) also as a second corresponding author, as he made an equal part of work for the writing and revision of this paper.

**GPR signature of Quaternary faulting: A study from the Mt. Pollino region, southern Apennines, Italy.**

Maurizio Ercoli[1-4], Daniele Cirillo[2-4], Cristina Pauselli[1-4], Harry M. Jol[3], Francesco Brozzetti[2-4]

[1]: Università degli Studi di Perugia, Dipartimento di Fisica e Geologia, Piazza dell'Università 1, 06123 Perugia, Italy.
[2]: Universita degli Studi "G. d'Annunzio" di Chieti-Pescara, DiSPUTer, via dei Vestini 31, 66100 Chieti, Italy.
[3]: University of Wisconsin - Eau Claire, Department of Geography and Anthropology, 105 Garfield Avenue, Eau Claire, WI, 54702.
[4]: CRUST Centro inteRUniversitario per l'analisi SismoTettonica tridimensionale, Italy.

Correspondence to: Maurizio Ercoli (maurizio.ercoli@unipg.it) and Daniele Cirillo (daniele.cirillo@unich.it)

**Abstract.** With the aim of unveiling evidence of Late Quaternary faulting, a series of ground penetrating radar (GPR) profiles were acquired across the southern portion of the Fosso della Valle-Campotenese normal fault (VCT) located at the Campotenese continental basin (Mt. Pollino region), in the southern Apennines active extensional belt (Italy). A set of forty-nine 300 MHz and 500 MHz GPR profiles, traced nearly perpendicular to this normal fault, were acquired and carefully processed through a customized workflow. The data interpretation allowed us to reconstruct a pseudo-3D model depicting the boundary between the Mesozoic bedrock and the sedimentary fill of the basin, which were in close proximity to the fault. Once reviewing and defining the GPR signature of faulting, we interpret near-surface alluvial and colluvial sediments dislocated by a set of conjugate (W- and E-dipping) discontinuities that penetrate inside the underlying Triassic dolostones. Close to the contact between the continental deposits and the bedrock, some buried scarps which offset wedge-shaped deposits are interpreted as coseismic ruptures, subsequently sealed by later deposits. Our pseudo-3D GPR dataset represented a good trade-off between a dense 3D-GPR volume and conventional 2D data, which normally requires a higher degree of subjectivity during the interpretation. We have so reconstructed a reliable subsurface fault pattern, discriminating master faults and a series of secondary splays. This contribution better characterizes active Quaternary faults in an area which falls within the Pollino seismic gap and is considered prone to severe surface faulting. Our results encourage further research at the study site, whilst we advise our workflow ideal also for similar regions characterized by high seismic hazard and scarcity of near-surface geophysical data.

**Key-words**: ground penetrating radar (GPR); Image processing; Faults; Neotectonics; Palaeoseismology; Earthquake hazards.

**1. Introduction**

A *"seismic gap"* is an area surrounded by regions struck by large earthquakes in historical or recent times. Such earthquake-free areas are characterized by the presence of seismogenic faults, whose past activity or possible quiescence is inferred on the basis of morpho-structural and/or paleoseismological data. The "*seismic gaps*" (McCann et al., 1979) show an apparent lack of historical seismicity but are candidate regions for the occurrence of large earthquakes in the near future (Mogi 1979; Plafker and Galloway 1989, Cinti et al., 1997, Galadini and Galli, 2003).
A recent example of a seismic gap "filled" by strong earthquakes is the Mt. Vettore area (central Apennines) during the 2016-2017 seismic sequence (Chiaraluce et al., 2017; Barchi et al., 2021 and references therein). Following the extensive coseismic ruptures mainly generated by the Mw = 6.5 "Norcia" mainshock (Villani et al., 2018; Brozzetti et al., 2019; Testa et al., 2019), this area is currently an ideal laboratory for many conventional and innovative geoscience disciplines and applications (e.g. Xu et al., 2017; Porreca et al., 2018; Brozzetti et al., 2020; Cirillo, 2020; Ferrario et al., 2018; Ercoli et al., 2020; Michele et al., 2020; Porreca et al., 2020; Buttinelli et al., 2021; Ferrarini et al., 2021; Pucci et al., 2021; Sapia et al., 2021; Villani et al., 2021). In fact, although the area being characterized by a complex alignment of normal faults, no important earthquakes were reported over the past ~1500 years before this seismic crisis (Cinti et al., 2019; Galli et al., 2019; Galli, 2020). Former geological and geomorphological studies suggested the possible occurrence of Quaternary faulting (Calamita et al., 1992; Brozzetti and Lavecchia 1994; Barchi et al., 2000), which was successively confirmed by paleoseismological (Galadini and Galli, 2003) and GPR surveying (Ercoli et al., 2013a; 2014). These studies revealed the occurrence of strong paleo-earthquakes and suggested that the Mt. Vettore master fault was "silent" but prone to cause future seismic events. However, invasive trenching due to complex logistics, environmental restrictions, high costs and the need for authorizations, cannot be applied systematically in many locations. Thus, Quaternary faults and associated basins characterized by an unsatisfactory definition of the seismotectonic framework have to be investigated with geophysical techniques. For all the above noted reasons, and since the Mt. Vettore case may represent an analogue of similar seismic gaps, the southernmost Apennines were studied through a dedicated research programme (Agreement INGV-DPC 2012-2013 and 2014-2015, Project S1 - Base-knowledge improvement for assessing the seismogenic potential of Italy, Brozzetti et al., 2015; Pauselli et al., 2015) aiming to improve the knowledge-base of seismogenic structures. In the research, focused also on the Calabrian region (Southern Italy) during the 2012-2015 period, structural geology, geophysical, and paleoseismological data were successfully acquired on the Mt. Pollino and Castrovillari fault systems (northern Calabria), providing evidence of Late Quaternary activity (Ercoli et al., 2013b; Cinti et al., 2015; Ercoli et al., 2015; Brozzetti et al., 2017b). This area, which is considered one of the most important seismic gaps in southern Italy, extends from the Mercure basin to the north until Campotenese basin and Castrovillari plain to the south, all characterized by Late Quaternary continental syn-tectonic sedimentation (Fig. 1a-c).

The paleoseismological trenching and radiocarbon dating document in the region the occurrence of paleo-earthquakes with $6.5 < M_w < 7.0$ and a recurrence time interval of ~ 1200 years (Cinti et al., 1997, 2002, 2015a,b; Michetti et al., 1997, 2000). But this high magnitude interval contrasts with the historical seismicity records, reporting a single significant $M_w$ 5.2 event occurred in 1693 (Tertulliani and Cucci, 2014). In the last three decade's instrumental seismicity recorded only two moderate seismic sequences climaxed in the Mw 5.6 Mercure (1998, September 9) and $M_w$ 5.2 Mormanno (2012, October 25) earthquakes. The latter occurred during a long-lasting sequence spanning the period 2010-2014, which included more than 6000 seismic events of $M_w > 1$ and activated at least three individual seismogenic sources (Passarelli et al., 2015; Brozzetti et al., 2017a, Fig. 1b). The gap between the low energy release, observed during the instrumented seismic sequences, and the high seismic potential estimated for the Quaternary faults, raised the question of whether even stronger earthquakes had shaken and could shake the area in the future. A recent and detailed parameterization of the Fosso della Valle-Campotenese fault (VCT in Fig. 1c) based on geo-structural and geomorphological mapping (Brozzetti et al., 2017a) as well as on seismological evidence (Totaro et al., 2014, 2015; Cirillo et., 2021), assesses a surface length of 15 km and a depth of at least ~10 km: the potential rupture-area is likely estimated to produce $M_w > 6.0$ earthquakes. As testified by earthquakes of the last century, such magnitudes, in the Apennines extensional belt generally produce coseismic surface faulting (e.g. Oddone, 1915; Pantosti and Valensise, 1990; Boncio et al., 2010; Brozzetti et al., 2019). However, Quaternary faulting for the VCT

structure is currently unclear, but geological and morpho-structural data suggest this fault has played an important role in determining the geometry and the recent sedimentary evolution of the basin.

The Campotenese basin and its VCT boundary fault is an example that summarizes the aforementioned issues: 1) lack of availability of paleoseismological data as the basin is entirely located within the Mt. Pollino National Park, thus requiring prior authorization from authorities; 2) lack of availability of publically accessible geophysical data; 3) no fresh recent surface displacements within the Holocene deposits have been observed along its trace. For all these reasons, the VCT represents an ideal case study suitable to test our working method.

We have conducted an explorative GPR field campaign across a VCT sector, suggested by discontinuous and smooth geomorphic scarps, as a screening tool for the definition of its possible Quaternary displacement history. The objectives of the paper are to: i) review and describe geophysical characteristics associated with a peculiar GPR signature of faulting, and propose a reference methodological workflow; ii) specifically check the efficiency of GPR prospecting to locate the VCT fault and to depict its subsurface pattern and spatial continuity at shallow depth; iii) provide new data to eventually relate the occurrence of $M_w > 6.0$ seismic events; iv) pave the way for other local geophysical studies and identify interesting sites for future ground-truthing and/or paleoseismological trenching; v) to have direct application and impact to the planning of future mitigation strategies for the reduction of surface faulting risk in the nearby urbanized areas.

**2. Tectonic setting and seismicity**

The Campotenese continental basin is located in the northernmost Calabria region south-west of the Mt. Pollino calcareous massif (southern Italy, Fig. 1). The bedrock of the basin consists of shallow water dolostones and limestones, Late Triassic to Middle Miocene in age, belonging to the Verbicaro tectonic unit (Ogniben, 1969; Amodio Morelli et al., 1976). It is generally referred to the western edge of the "Apenninic Platform", a thick (> 4 km) carbonate shelf, that underwent compression during the Middle-Late Miocene times and was translated over an eastern basinal domain (Lagronegro-Molise basin; Patacca and Scandone, 2007; Vezzani et al., 2010 and references therein). From the bottom to the top, the bedrock succession includes late Triassic dolostones, Cretaceous limestones, and Paleocenic-Lower Miocenic calcarenites cross-cut by the pillow lava basalts belonging to Liguride units of the northern sector of Calabrian arc (Quitzow, 1935; Grandjaquet and Grandjaquet, 1962; Amodio Morelli et al., 1976; Ghisetti and Vezzani, 1983; Iannace et al., 2004, 2005 and 2007; Liberi et al., 2006; Filice et al., 2015; Tangari et al., 2018).

The origin of the Campotenese basin, however, is related to a set of NW-SE striking extensional faults which, during the Middle-Late Pleistocene, displaced the contractional tectonic pile, favoring the deposition of alluvial and lacustrine sediments in a subsiding intra-mountain depression (Servizio Geologico d'Italia 1970). This set of conjugate SW- and NE-dipping normal faults represents the local expression of the Quaternary extensional belt that develops all along the Italian peninsula, nearly parallel to the axial zone of the Apennines, from northern Tuscany to the Calabrian Arc (Brozzetti 2011). North of Campotenese, (Lucania and southern Campania) the Apennine extensional belt includes several continental basins and their boundary faults, as the Irpinia, Vallo di Diano, Tanagro, Melandro-Pergola and Val d'Agri (Ascione et al., 1992; Maschio et al., 2005; Amicucci et al., 2008; Villani and Pierdominici, 2010; Brozzetti, 2011; Filice and Seeber, 2019; Bello et al., 2021). To the south, it continues with the Crati graben that dissects the northern sector of the Calabrian Arc (Tortorici et al., 1995; Brozzetti et al., 2017b).

On the regional scale, the Quaternary normal fault array controls the release of major seismicity, as suggested by the distribution of supra-crustal instrumental earthquakes (INGV, 2020 and ISIDe, 2007) and of the strongest historical events (Fig. 1a, Tertulliani and Cucci, 2014; Rovida et al., 2020). The recent seismic activity as well as paleo-seismological investigations claim that most of the faults bounding the Quaternary basins are seismogenic and therefore enable, in some cases, to associate major past earthquakes with specific structures (e.g. Pantosti and Valensise, 1990; Cello et al., 2003; Spina et al., 2009; Brozzetti et al., 2009; Villani and Pierdominici, 2010). These same studies highlight that the kinematics of the Quaternary faults and the focal mechanisms of the major earthquakes are mutually consistent and are mainly compatible with an SW-NE direction of extension (RCMT and TDMT databases by Pondrelli, 2006 and Scogliamiglio et al., 2006). Other authors have recognized in the surrounding regions an oblique normal-lateral faults kinematics (e.g. Rossano and Sybaris faults, Galadini et al., 2001, Cinti et al., 2015b). The fault investigated in this work has been pointed out in more detail by Brozzetti et al. (2017a) in the frame of a larger study focussed on the Quaternary and active faults at the Calabrian-Lucanian boundary (Fig. 1a). In the region, three main sets of normal faults, with prevailing dip-slip kinematics, have been mapped: a western one, consisting of E- to NNE-dipping faults (red lines in Fig. 1b), and two other main sets of W-to SW-dipping fault segments (dark-blue and blue lines in Fig.1b). The Rotonda-Campotenese Set (ROCS) is a right-stepping en-échelon master fault developed for a total length of 15 km with an average N160E strike (blue, yellow rimmed lines in Fig.1b). ROCS is composed by two fault segments: i) the westernmost Fosso della Valle-Campotenese fault (VCT), which extends from the southern border of the Mercure basin to the SW boundary of the Campotenese basin, and ii) the Rotonda-Sambucoso fault (RSB), which branches-out from the VCT segment in the central part of the ROCS. In the northern sector, the two segments are averagely spaced ~ 2.5 km at surface and linked at a depth of ~ 9-10 km (Cirillo et al., 2021), cross-cutting the middle-Pleistocene ~ E-W striking Cozzo Vardo-Cozzo Nisco fault (CVN, light-blue line in Fig. 1b). Along the E-side of the Campotenese basin, the VCT is generally buried by Holocene deposits, but its location can be inferred based on stratigraphic observations and geomorphic features, such as sharp ridge fronts, linear scarps, and slope breaks. The VCT controls the distribution and thickness of the clastic fill basin (Middle Pleistocene-Holocene in age, according to Schiattarella et al., 1994) that reaches the maximum thickness (~ 30 m) in the western sector (VCT hanging wall, see boreholes stratigraphy at http://sgi2.isprambiente.it/mapviewer/). The spatial relationships, at surface and depth, between the Quaternary fault segments and the hypocenters of the re-located 2010-2014 seismic events (Totaro et al., 2015; Brozzetti et al., 2017a; Napolitano et al., 2020, 2021; Pastori et al., 2021) suggest that the VCT is a good candidate as a seismogenic source for the Mw 5.2 (2012, October 25) Mormanno earthquake, as well as for strong paleo-events.

FIGURE 1 HERE

**3. Methodology**

Ground penetrating radar (GPR) is a high-resolution geophysical method able to provide detailed images of the shallow sub-surface. This methodology is based on the recording of EM reflections, with operative frequencies for geoscience applications generally between 10 MHz and 1000 MHz, depending on the transmitting and receiving antennae. The GPR reflections rise from dielectric permittivity contrasts between the subsurface targets and the surrounding media, which in geological and archaeological applications typically correspond to geo-lithological changes or water content variations (Jol, 2009). In low conductivity materials ("low-loss"), the maximum investigation depth is generally comprised within few tens of meters (Davis and Annan, 1989). The latter is however controlled also by the electrical conductivity, which for high values causes radar signal attenuation (Annan, 2001). The reflections are recorded as a function of the Two-Way-Travel time (TWT) propagation, and displayed as a 1D GPR trace. Several GPR traces displayed along a transect build-up a radar profile or "radargram", that is the 2D representation of the GPR reflections, more commonly identified as the conventional GPR output. A GPR dataset may be provided also as a 3D volume, which has been common for 25+ years in research applications and recently more widespread due to a wider diffusion of commercial GPR instruments equipped with arrays of antennae. The GPR is used in many research and applied fields, such as geological, sedimentological, geomorphic, hydrogeological applications (Bristow and Jol, 2003; Jol, 2009), and also in archaeological and engineering studies (Conyers, 2016; Daniels, 2004; Goodman and Piro, 2013; Utsi, 2017). In active tectonic context, several 2D/3D GPR studies have already imaged buried tectonic structures. These studies have shown geophysical images of faulting, supporting and/or extending outcrop, borehole, trench data, and contributing to base-knowledge of seismogenic structures as well as to the seismic hazard assessment of several regions around the world. Among the pioneers, we can mention Benson (1995), Smith and Jol (1995), Busby and Merritt (1999), Cai et al. (1996) and Liner and Liner (1997), and on the successive twenty years, other 2D GPR studies were achieved across several faults worldwide (Audru et al., 2001; Demanet et al., 2001; Overgaard and Jakobsen, 2001; Bano et al., 2002; Liberty et al., 2003; Reiss et al., 2003; Slater and Niemi, 2003; Malik et al., 2007; Wallace et al., 2010; Yalciner et al., 2013; Imposa et al., 2015; Anchuela et al., 2016; Nobes et al., 2016; Matos et al., 2017; Pousse-Beltran et al., 2018; Zajc et al., 2018; Zhang et al., 2019 and Shaikh et al., 2020). In Italy, only a few GPR studies are currently available across normal faults (e.g. Salvi et al., 2003; Jewell and Bristow, 2006; Pauselli et al., 2010; Roberts et al., 2010; Ercoli et al., 2013a; 2014; Bubeck et al., 2015; Cinti et al., 2015). Over time, 2D GPR acquisitions were flanked by an increasing number of pseudo-3D or full-3D GPR studies (Grasmueck et al., 2005). Grasmueck and Green (1996) traced the future path of three-dimensional GPR applications, providing a dense 3D GPR volume to image fractures in a Swiss quarry. The study opened the possibility to three-dimensional GPR imaging of subsurface geological structures. Successive studies extended the approach to characterize active faults in different tectonic regimes combining 2D and pseudo-3D GPR surveys (e.g. Gross et al., 2002, 2003, 2004; Green et al., 2003; Tronicke et al., 2006; McClymont et al., 2008, 2009, 2010; Vanneste et al., 2008; Christie et al., 2009; Carpentier et al., 2012a,b; Malik et al., 2012; Brandes et al., 2018). A review of the near-surface GPR faulting studies suggests some reflection characteristics as possible indicators for the detection of subsurface fractures and faults (e.g. Smith and Jol, 1995; Liner and Liner, 1997; Reiss et al., 2003; Gross et al., 2004; McClymont et al., 2008, 2010 and Bubeck et al., 2015). Among these, sharp lateral reflectivity variations, interruptions of the reflections, and the presence of hyperbolic diffractions are considered convincing evidence, as shown also by numerical simulations (Ercoli et al., 2013a; Bricheva et al., 2021). In addition, we have accounted for additional GPR indicators identified for Quaternary faulting in similar environments (Ercoli et al., 2013a,b; 2014; 2015), which are linked to the geometry of stratigraphic deposits across fault zones: i) reflections abrupt truncating and offsetting along sub-vertical discontinuities (especially in the case of a normal fault); ii) reflection packages thickening as they approach the fault strands; iii) abrupt lateral dip variation of the reflections; iv) peculiar reflection package geometries, with contorted reflection patterns resembling *"colluvial wedges"*, which McCalpin (2009) defines as deposit due to *"subsidence and sedimentation of the hangingwall and erosion of the morphological scarp in the footwall"*; v) localized strong GPR signal attenuation due to the presence of conductive media within the main fault zone.

Based on the research and criteria reviewed above, we carried out a near-surface interpretation of faulting based on the co-existence of most of these features along several adjacents analyzed GPR profiles. These conditions strengthen the interpretation of each profile and aids to highlight the spatial continuity of the interpreted structures over linear distances of at least many tens, or hundreds, of meters.

*3.1 GPR and GNSS survey*

Three different geophysical field campaigns carried-out during the 2014-2015 years, a dataset of 49 GPR profiles was acquired in the southern sector of the ROCS across the VCT fault segment (Fig. 1b-c), covering a buffer zone of ~

400 m and ~ 200 m respectively along and across the fault strike (area of ~ 8 Ha), for a total linear length of GPR

profile about 4100 m collected using a Common Offset (CO) configuration (Fig. 2).

FIGURE 2 HERE

We used a Zond 12e GPR system equipped with 300 and 500 MHz antennae. The lower frequency antennae was ultimately preferred and considered the best trade-off between maximum resolution and achievable signal penetration (in our case ~ 4 m) concerning the surveyed materials and wanted subsurface structures. The GPR was equipped with an odometer wheel to measure the radar profiles' length and with a Topcon GR-5 Global Navigation Satellite System (GNSS) receiver to achieve accurate positioning of GPR traces and profile. Considering the scarce presence of obstacles across the survey site and the good satellite coverage, we opted for a Network Real-Time Kinematic positioning (NRTK, connected to the NETGEO network), measuring coordinates and elevations with centimetre accuracy, and stored directly within the SEG-Y GPR files.

Three datasets were acquired after preliminary fieldwork and collection of geological structural data at the surface and which allowed us to infer the possible location of the fault trace. The average NE-SW direction of the GPR lines was initially planned with the primary purpose of intersecting the VCT fault ~perpendicularly to its SW-NE strike, as reported by literature and visible by surface evidence. This solution theoretically allows a more reliable interpretation of the investigated structure by reducing the effect of the apparent dip-direction and dip-angle of both stratifications and faults.

The acquisitions carried out in 2014, first resulted in twelve SW-NE GPR profiles collected in the southern sector of the basin (CMT light-blue lines in Fig. 2a), which was a flat land characterized by Quaternary alluvium. The second acquisition encompassed four additional radar profiles collected in the same area, and another nine radar profiles progressively moving to north, which were collected with slightly different and converging orientations in the central sector (CMT green lines Fig. 2a). This solution was pursued for two main reasons: 1) to avoid directly surveying the outcropping dolostones (only partially crossed with two northernmost profiles) characterizing two hills *h1* and *h2*

(dashed white polygons in Fig. 2), and thus focussing only on the sedimentary cover which is our target for possible

Quaternary faulting; 2) to optimize, through a preliminary GPR data interpretation, the future acquisition schemes by figuring out the dip direction of the buried geologic structures of interest. In fact, similarly to the interpretation of reflection seismic profiles, the "apparent dip" of reflections in bidimentional radar profiles should be considered to achieve a reliable 3D conceptual model.

[revised manuscript text omitted]
). The quality of the radar reflections and the remarkable depth reached (~ 6 m, Fig. 6b) suggest this rock type is an excellent dielectric medium (corresponding to higher frequency content zone in the 2D spectrum of Fig. 6c). However, its reflection pattern is not spatially homogenous, being characterized by oblique and sub-parallel reflections. The latter are interpretable as dolostone beds of moderate (25-30°) W and E "apparent" dip on the respective sides of the surveyed dolostone hills. In addition, these reflections are frequently cut and slightly displaced by apparent high-angle (60-65°) phase discontinuities, highlighted by a dense hyperbolic diffractions pattern (radar profile cmt2n, Fig. 7a), suggesting intense fracturing and little faults displacing the dolostone (Fig. 7b). This radar signature was recorded not only in correspondence of the outcropping carbonate but also in the transition slope areas covered just by a thin soil layer (Figs. 7b,c). In the southern side of *h1*, an outcrop with thin microbialitic laminae allows one to measure the attitude of the bedding (NNW dip, ~ 30-35° dip angle) as well as two sets of major and minor joints (SW and SE dip and dip angle of ~ 40-45°, respectively) fitting with GPR reflections.

Apart from its internal heterogeneities, the GPR signature of the Triassic dolostones can be considered as a well- defined depositional facies (*fc1*) (Sangree and Widmier, 1979; Huggenberger, 1993; Beres et al., 1999; Jol and

Bristow, 2003). A different radar signature *fc2* is defined for the profile sectors on the sides of *fc1*. This second facies is characterized by prominent laterally-continuous and sub-parallel reflections in the very shallow depth range (< 1 m, just beneath the direct arrivals), stratigraphically sealing underlying reflections 1-3 m deep: the latter are more discontinuous, wavy, and contorted, with moderate to low reflectivity and encompassing sparse diffraction hyperbolas (in unmigrated data, Fig. 7a). This reflection pattern onlaps onto a generally prominent wavy reflection (Fig. 7a,b), which typically marks the transition to strong signal attenuation deeper in the section.

FIGURE 7 HERE

The reflection package belonging to *fc2* corresponds to the alluvial/colluvial deposits (Fig. 7b-d), outcropping on the sectors with flat topography, which represent the GPR profile sectors we've carefully inspected to find for geophysical evidence of Quaternary faulting. A key-layer for this research is the described prominent, wavy reflection, as it can be recognized in many radar profiles. The related interpretation is not straightforward in the absence of direct data (e.g. boreholes and/or paleoseismological trenches) or at least without additional geophysical data. A strong GPR

reflection suggests significant variation of the dielectric constant between the two media so that most of the incident energy is reflected back to the receiver at the surface. This wave behaviour is potentially explained by several geological models, such as: i) a high dielectric contrast may be a result of a sharp soil moisture variation (Ercoli et al.,

2018); ii) a sharp erosional, stratigraphic or tectonic boundary within heterogeneous deposits (Ercoli et al., 2015), or iii) a contact between two considerably different lithologies, such as unconsolidated deposits laying above a bedrock substrate reflecting back all (or almost all) the incident signal (e.g., Frigeri and Ercoli, 2020). In addition, the possible role of conductive deposits (e.g. high clay content) should not be discounted to explain the occurrence of strong attenuation. Several considerations are at the basis of the GPR data interpretation:

1) the available well logs show the Pleistocene-Holocene alluvium and colluvium layered above the carbonate bedrock

~20-30 m depth (Brozzetti et al., 2017a), a greater depth than the strong GPR reflection. However, it should be observed that the area drilled is located ~2.5 km away on the north-westernmost sector, over the depocenter of the

Campotenese basin, whereas the studied GPR site is placed just on its eastern border, in proximity to emerged dolostone hills; 2) only terraced Middle-Pleistocene silts and sands (Schiattarella et al., 1994) and slight coatings of

Late Pleistocene colluvium (generally < 2 m thick) are documented to outcrop in the eastern sector of the basin (footwall of VCT fault) (see Fig. 7 in Brozzetti et al., 2017a); 3) the subsurface geometries highlighted by the prominent GPR reflection and underlying reflection pattern suggest a relatively thin layer of sedimentary deposits resting on a fractured substratum. Its top surface is progressively deepening towards the W, thus providing increased space for settling sediments and thus a gradual thickening of deposits is observed from E to W.

In light of the above considerations, we interpret the prominent, wavy GPR reflection as a buried top layer of carbonate (e.g. as observed e.g. by Bubeck et al., 2015), in our case represented by the Triassic dolostone formation. The latter is lying at a shallower depth (1-3 m) beneath shallow and poorly consolidated Quaternary deposits, across both sides of the surveyed hills. Thus, after picking such a prominent reflection event on all the radar profiles, the top of bedrock surface was reconstructed as shown in Fig. 8a (coloured surface). In this figure, we display also an overlay of a recent structural map of the basin (modified after Brozzetti et al., 2017a) reporting the area dissected by a set of en-echelon fault splays to the West associated to the VTC segment. Thus, analyzing the geophysical characteristics of the prominent, wavy reflection in terms of a structural interpretation, the main peculiar characteristic is the clear "stepped" geometry of some sectors (Figs. 5b, 6b, 7b, 8b), namely breaks of its continuity associated to lateral sharp variations of depth (linked to sediment growth and onlaps). We also notice other geophysical features, which can be observed in the stratigraphy of overlying deposits (*fc2*): some reflections are semi-continuous to discontinous (sharp variation in signal amplitude and phase) and display evident lateral variation of the dip angle.

FIGURE 8 HERE

These broken reflection packages present truncantions (e.g. Smith and Jol, 1995), displacements, and hyperbolic diffraction events (insets of Fig. 8b1,b2). Such peculiar GPR signature is therefore compatible with coseismic displacement due to Late Quaternary surface faulting events (Fig. 8b). Contorted reflections across the main discontinuities frequently show localized strong attenuation of GPR signal (Fig. 8b). The attenuation might be linked to their high dip-angle, causing a minor amount of energy being reflected back to the antenna, but, more likely, due to the presence of conductive fine soils nearby faulted zones (e.g. circle 1 in Fig. 8b). These conditions can be linked to different depositional facies across fault zones (McClymont et al., 2010) e.g. including colluvial wedges (Reiss et al., 2013; Bubeck et al., 2015) or deposits deriving from degradation of fault scarps (detailed interpretation within the caption of Fig. 8b). Using all such stratigraphic evidence and geophysical markers of faulting, we have therefore interpreted and classified synthetic (W-dipping, blue) and antithetic (E-dipping, red) normal faulting events (Fig. 8b). During the interpretation process, the faults were picked using solid lines (fault sticks); when the presence of geophysical markers of faulting were uncertain, a dashed fault segment was initially added and revised a second time their possible connection among nearby GPR profiles.

The interpreted faults present a dip angle between 65-75° and variable amount of displacement (D), estimated by correlating the position of the top of the carbonate substratum in the footwall and hanging wall blocks (e.g. scheme summarized in the inset of Fig. 8b3). Considering the GPR profile of Fig. 8 as representative for the studied VCT sector, D is not exceeding ~1 m for the W-dipping splays within the Quaternary sediments (~ 0.5 m for the E-dipping splay). A displacement D of ~1.5 m was derived across the sharp boundary between the Triassic dolostone and the Quaternary deposits (easternmost fault on Fig. 8b), being interpreted as the main fault. This clear contact is characterized in all profiles by hyperbolic diffractions (in unmigrated data), variable dip angle, abrupt truncations, sharp lateral variation of the reflectivity suggesting a wide fault zone (Figs. 3 to 9), controlling the above mentioned Quaternary splays. By interpolating all the fault sticks placed in adjacent profiles, we have created the fault surfaces, that show a good degree of continuity from north to south (Fig. 9). For the studied sector of the VCT, we have reconstructed the tridimensional fault-network and the geometry of the associated synsedimentary deposits at a metric scale of observation (Fig. 9).

**5. Discussion**

**5.1 Inferences from subsurface 3D model**

The perspective view of Fig. 9a shows a 3D structural scheme of the main tectonic lineaments at the basin scale displaying a NW-SE faults strike (modified after Brozzetti et al., 2017a) in relation to the GPR investigated area (white rectangle). Our GPR interpretation enriches many of the details such a former structural scheme across the southern VCT segment. We highlight an en-echelon system of two main SW and NE-dipping faults as well an articulated set of extensional meso-faults within the Quaternary sediments. The high-angle GPR discontinuities identified in the study (e.g., Fig. 9b) show a considerable continuity in the NW-SE direction (accurate 3D structural recontruction in Fig. 9c), dissecting not only Quaternary alluvial-colluvial deposits (except for the very shallow *fc2* layers), but also deeper stratigraphic layers.

The reconstructed faults mark a horst-graben structure, mostly buried within the Campotenese basin, which locally emerges from the Quaternary deposits. In the investigated area it corresponds to a NNW-SSE elongated topographic high (*h1* and *h2* in Fig. 2a) made by the Triassic dolostone. This horst is bordered toward the W and towards the E by SW- and NE- dipping normal faults, respectively (Figs. 9b, c). In Fig. 9c, the fault-set *d1,* together with its antithetic set *d2,* shows the maximum displacement and the most evident deformation of the adjacent sub-surface deposits. The variations of thickness of such Quaternary deposits are consistent with the horst and graben configuration. Thinning is observed in correspondence with the raised buried blocks, whereas thickening, wedge-shaped as well as chaotic geometries correspond to the lowered blocks. The main fault of set *d1* can be considered a conjugate fault of the VCT (Fig. 8b), separated by a right step-over of about 0.5 km from the segment that borders the eastern basin (Figs. 2c, 8a). Thus, also the fault-set *d3* and *d4* located on the eastern part of *h1* and *h2*, can be hierarchically classified as a network of minor splays embedded in the southern junction zone between the two VCT segments (Fig. 9c).

The three-dimensional model (Fig. 9a,c) highlights that these faults, despite having a typical Apenninic NW-SE trend, are characterized by a complex polymodal pattern of strikes, with alternating N-S to NW-SE direction. Therefore, such a polymodal character which was observed along all the extensional structures of the area (Brozzetti et al., 2017a) is also confirmed at the GPR scale along this VCT sector. A dedicated statistical analysis of the reconstructed fault planes is reported in the stereo plots of Fig. 9d (d1-d3= W-dipping faults; d2-d4= E-dipping faults).

FIGURE 9 HERE

**5.2 Seismic hazard implications**

The combination of geological and seismological data may suggest outcropping Quaternary faults being capable of releasing earthquakes, but the determination of the maximum expected magnitude along these faults might not always be well constrained. An estimate can be made using well-known scale-relationships (Wells and Coppersmith, 1994; Wesnousky et al., 2008; Leonard, 2010; Stirling et al., 2013) with knowledge of the geometric parameters (e.g. fault length, area and depth), which are often difficult to assess. These scale relations can also be applied also to Quaternary scarps originated by cumulative coseismic faulting produced by medium-strong earthquakes (generally M > 6). Nevertheless, only through paleoseismological analysis, by sampling and dating the stratigraphy at different levels, is it possible to date and distinguishing the amount of slip of each seismic event. But in cases like the VCT, the GPR data assume a key-value since provided key fault parameters where no direct information on the nature of the surveyed deposits and no accurate dating is available. Our GPR interpretation by itself doesn't allow one to extract any date for a single earthquake, nor identify a succession of past seismic events and neither establish recurrence times (Galli, 2020). However, it confirms a segmentation of the VCT and the presence of buried splays, which appear to have exerted a strong control on the deposition of Late Quaternary sediments. The location of Quaternary ruptures at a shallow depth in a flat land of an intra-mountain basin presently undergoing alluvial and colluvial sedimentation, suggests their occurrence might be attributed to the Holocene. Thus, pointing out normal faulting of Holocene deposits would be, in itself, a very important and novel result for the Campotenese area. A Middle-Late Pleistocene age of activity was suggested for the Mercure and Campotenese boundary faults by Schiattarella et al. (1994) and Brozzetti et al. (2017a), with Holocene activity indirectly inferred on the base of morpho-structural observations. More recently, an earthquake-structure association with the recent 2010-2014 Pollino seismic sequence has been reconstructed through cross-sections and relocated seismicity in Cirillo et al., (2021).

Our data are promising because the GPR facies interpretation highlights the possible presence of small-scale grabens or half-graben (maximum estimated fault zone width of ~ 160-170 m, inset c1 of Fig.9c) and the likely fault-related deposits (e.g. as observed by Reiss et al., 2013 and Bubeck et al., 2015) at shallow depth. This inference would testify to not only the persistence of extensional deformations up to the Late Quaternary but would even imply the occurrence of episodes of surface faulting. In other words, the Campotenese basin may have been affected in the relatively recent past by medium-strong earthquakes, larger than the 2010-2014 mainshocks. It should be in fact considered that historical events with $6 < M_w < 7$ sourrounded the area, being documented a further ~ 50 km north (1857 - $M_w$ 7.1) and ~ 60 km south (1184 - $M_w$ 6.7, Fig. 1a) (Rovida et al., 2020). Some paleoseismological earthquakes with inferred magnitude $6.5 < Mw < 7$ are attributed to the Castrovillari fault, located ~ 20 km SE and also falling within the Pollino seismic gap (Cinti et al., 1997; Michetti et al., 1997; Cinti et al., 2002, 2015).

The estimates of the VCT fault-length provide an overall value of 15 km (Brozzetti et al., 2017a) which is compatible, in the case of a complete rupture, with the maximum expected magnitudes of Mw = 6.45 (Wells and Coppersmith, 1994) and Mw = 6.8 (Wesnousky et al., 2008 and Leonard, 2010), therefore well capable to produce surface breaks. Being the source of the most recent earthquakes (2012 - $M_w$ 5.2; 1894 - $M_w$ 5.1; 1708 - $M_w$ 5.8 and perhaps 1693 - $M_w$ 5.2) affecting the study area estimated at ~ 8 km depth (Totaro et al., 2015; Brozzetti et al., 2017a; Napolitano et al., 2020, 2021, Sketsiou et al., 2021), the level of seismic energy released by such historical seismic events would likely be not enough to generate the VCT ruptures at surface. Therefore, it sounds reasonable that the hypothesis of past earthquakes occurrence, nucleated from the VCT, with a magnitude sufficiently high to cause the buried coseismic ruptures, highlighted by our GPR interpretation, which were then subsequently erased at surface by footwall erosion and sedimentation at the hanging wall. In addition, because historical catalogs do not show events with $M_w > 6$ (Rovida et al., 2020), a very energetic earthquake could have likely occurred before the period covered by the available seismological catalogs, proving new perspectives on the actual seismic hazard of the area.

**6. Conclusions**

Our novel GPR data and dedicated workflow allowed us to obtain a detailed 3D model of the southern sector of the Fosso della Valle - Campotenese fault (VCT) in the continental Campotenese basin, a seismic gap in the Mt. Pollino region (Southern Italy). The processing, analysis, assemblage, and interpretation of the 49 GPR profiles was pursued using expertise, techniques, and tools borrowed from seismic reflection industry applications. The non-destructive GPR survey did not require special authorizations and was relatively fast and low cost. The pseudo-3D configuration was an efficient compromise between spatial coverage and duration of the data acquisition (four days of fieldwork). On the other hand, the data processing was non-trivial, requiring about six months overall to set up an optimized workflow, due to challenging data characteristics, such as the steep and rugged topography and the sharp lateral variations of dielectrical properties of media (Triassic Dolostones vs Quaternary deposits).

Our structural reconstruction derived by GPR data interpretation shows several sets of sub-vertical discontinuities within the near-surface (~ 1-4 m depth), which we interpreted as a pattern of extensional surface faulting. Such faults are bounding small local "graben or semi-graben-like" structures, which cut an hypothesized Holocene age clastic cover and underlying Triassic dolostones. We have also identified some chaotic and laterally discontinuous GPR-stratigraphic facies, interpreted as near-fault post-earthquake deposits (i.e. colluvial wedges ?). These shallow structures suggest the possibility that surface faulting due to past strong earthquakes ($6 < M_w < 7$) occurred in relatively recent times in the study area. Its traces at surface were possibly later levelled by the concurrent natural processes of erosion, aggradation and, anthropogenic activities. As our results confirms the presence of seismic potential and thus the possible occurrence of a large earthquake in the future, we wish the primary effect of our study to be one of raising the level of attention regarding the seismic hazard in the Campotenese area, as well as prompting further research. Upon ground truthing, our work may represent a preparatory study for further geophysical surveys (3D GPR and other methods), as well as direct analysis including trenching, drilling, sampling campaigns and dating (e.g., luminescence, radiocarbon, etc). Although a further multidisciplinary approach would be necessary to achieve a quantitative (i.e. slip rates and recurrence times) assessment of the seismogenic potential of the study area, we firmly promote, particularly where near-surface data is lacking, a widespread use of the presented GPR workflow on other seismic gaps worldwide.

*Author contributions*. ME and DC contributed equally to this work as first authors. ME, DC, CP, FB led the fieldworks. ME analyzed, processed the GPR and GNSS data. ME, DC, CP, HMJ, FB contributed to the paper conceptualization and writing. ME and DC managed all data in the GIS environment and within 3D interpretation programs (OpendTect, Move), as well as they have created all the figures. DC realized the final 3D structural-geological model through Move software. All authors reviewed and edited all the drafts.

*Competing interests.* The authors declare that they have no conflict of interest.

*Disclaimer. Publisher's note:* Copernicus Publications remains neutral with regard to jurisdictional claims in published maps and institutional affiliations.

*Special issue statement.* This article is part of the special issue "Tools, data and models for 3-D seismotectonics: Italy as a key natural laboratory".

**Acknowledgments**

We sincerely thank Leonardo Speziali, Prof. Costanzo Federico, and Roberto Volpe for their support during the field operations, as well as Khayal Gavarof for his massive and valuable collaboration in data organization and processing. We thank QGIS (https://www.qgis.org/it/site/) for providing the software with an open-source license, Petroleum Experts (https://www.petex.com/products/move-suite/), and dGB (https://www.dgbes.com/) for providing the academic licenses MOVE and OpenDtect software. We acknowledge NETGEO for academic access to the NRTK network (http://www.netgeo.it/). We would also like to thank the Ministero dell'Ambiente e della Tutela del Territorio e del Mare (MATTM) and the Regione Calabria for providing free access to geospatial data such as DTM and aerials (Regione Calabria - www.regione.calabria.it, under license IODL 2.0. - https://www.dati.gov.it/iodl/2.0/). We sincerely thank also the editor Luca de Siena and two anonymous reviewers for their constructive corrections and suggestions which contributed to improve our manuscript. The paper is the result of collaboration within the framework of the Interuniversity Center for 3D Seismotectonics with territorial applications - CRUST (https://www.crust.unich.it/). The GPR dataset presented in this study is available on request from the corresponding author.

*Financial support.* The study has benefited from several funding sources including: Agreement INGV-DPC 2012-2013 & 2014-2015, Project S1 – Miglioramento delle conoscenze per la definizione del potenziale sismogenetico - Base-knowledge improvement for assessing the seismogenic potential of Italy, https://sites.google.com/site/ingvdpcprojects1/home, resp. Cristina Pauselli, funded by Italian Presidenza del Consiglio dei Ministri – Dipartimento della Protezione Civile (DPC). The paper does not necessarily represent DPC official opinion and policy.

Review statement. This paper was edited by Luca De Siena and reviewed by two anonymous referee.

[revised manuscript text omitted]

**Figures and Tables captions:**

**Figure 1 - Location maps of the study site (DTM sources: TINITALY by Tarquini et al., 2012 and by Regione Calabria - www.regione.calabria.it, under license IODL 2.0. - https://www.dati.gov.it/iodl/2.0/): a) the image illustrates the southern Italian peninsula with the regional faults pattern and the historical strong earthquakes (Rovida et al. 2020); b) map showing the studied region with local faults (modified after Brozzetti et al. 2017a), and epicenters (stars) and focal mechanisms of the mainshocks of the 2012-2014 seismic sequence (Scognamiglio et al., 2006); c) location of the GPR survey area within the Campotenese Quaternary basin crossing the Fosso della Valle - Campotenese (VCT) fault.**

**Figure 2 - GPR acquisition campaigns: a) GPR profiles collected at the study site Campotenese ("cmt", where "n" and "s" stay for North and South, "h1" and "h2" indicate the two Dolostone hills outcropping in the basin) during the three field visits (aerial image source: Regione Calabria - www.regione.calabria.it, under license IODL 2.0. - https://www.dati.gov.it/iodl/2.0/); b) acquisition phase using the 300 and 500 MHz antennae (in the insert) and GNSS receivers used for accurate data positioning; c) GNSS base station set up during the fieldwork.**

**Figure 3: Topographic correction of GPR profiles: a) example of accuracy degradation of GNSS data, displaying an outlier both in map view and in topographic profile, on which the positioning error is considerable; b) GNSS coordinates and topographic profile after the correction; c) raw GPR section displaying high reflectivity in the central sector; d) example of full processed profile with topography displaying various reflection patterns encompassing dipping reflections and diffractions. Vertical exaggeration is 4.**

**Figure 4: Migration tests performed during the GPR data processing: a) unmigrated 2D GPR profile, 300 MHz antennae, displaying hyperbolic diffractions (white arrows); b) migrated profile using a constant velocity v = 0.07 m/ns, light-blue arrows indicate good diffractions collapse; c) migration output obtained with a constant velocity v = 0.09 m/ns, with dark-blue arrows suggesting good migration results (migration artefacts are shown by red arrows); d) migration results using a constant velocity v = 0.11 m/ns, with dark-blue arrows highlighting good hyperbolas collapse, particularly within the high reflective unit; red arrows highlight clear migration smiles.**

**Figure 5: Example of 2D time-migration of radar profiles: a) example of hyperbolic diffractions fitting used for 2D velocity model building; a constant velocity value (0.07 m/ns) was assumed in deeper no-diffraction areas for interpolation purposes; b) 2D time-migration results, highlighting the good performance of the process, which collapsed the hyperbolic diffractions (white arrows) and restored reliable reflection geometry.**

**Figure 6: GPR data visualization: a) fence diagram showing the three-dimensional location of some representative GPR profiles in the northern sector of the study site; b) bidimensional GPR profile (cmt3n, see figure 2a for location) displaying the central high reflective sector and dipping reflections across the hill; c) spatial variation of a 2D amplitude-frequency spectrum linked to variable physical properties of media along the profile cmt3n. Vertical exaggeration is 4.**

**Figure 7: Correlation between GPR profiles and outcropping geology at the study site: a) unmigrated 300 MHz profile (cmt2n, see fig. 2b for location) displaying numerous hyperbolic diffractions; b) migrated profile displaying the apparent dip associated to fractured dolostone formation (facies fc1) and Quaternary deposits in the attenuated sectors (GPR facies fc2); c) Quaternary deposits of the basin (on the background) surrounding the Triassic Dolostone formation outcropping on the hill h1 . The yellow arrows indicate the bedding, such as the stereo-net (left-side inset); the right-side inset report a detail of the laminae visible on site and nearby; d) an example of Quaternary colluvial and alluvial deposits outcropping nearby the survey site. Vertical exaggeration is 2.5.**

**Figure 8: GPR data interpretation: a) three-dimensional image of the surveyed area (see fig. 1c for location), displaying the Dolostone outcrops (grey colour). Blue dashed lines are the VCT and RSB faults (fig. 1b), whilst the light blue is CVN fault. In yellow lines the GPR profiles; the coloured surface is the interpreted Dolostone top reflection (DTM source: Regione Calabria - www.regione.calabria.it, under license IODL 2.0. - https://www.dati.gov.it/iodl/2.0/); b) migrated radar profile with the main interpreted normal faults (blue and red are W- and E- dipping structures, respectively) as well as related sedimentary structures within the Quaternary deposits (unmigrated data in b1 and b2); the inset b3 is a schemathic representation illustrating the methodology used for extraction of the GPR fault displacement (D: displacement; T: throw; H: heave). GPR facies fc2 shows semi-continuous and sub-horizontal reflections (Quaternary deposits) onlapping fc1 (Triassic Dolostones, black line is the "top"). In circle 1: reflections package thickening and truncation with localized attenuation are likely interpretable as "colluvial-wedge-like" (cw?) features, or deposits from degradation of earthquake fault free-face nearby of the hanging-wall (D ~ 0.6 m). In circle 2: _fc2_ show more discontinuous, from subparallel to wavy reflections package downlapping the lower top Dolostone; the**

asymmetric, truncated reflections thickening is bounded by two conjugate normal fault strands (east dip D ~ 0.5 m, west-dip D = 0.4 m) displacing both fc1 and fc2. In circle 3: contorted reflections package with limited continuity, displaying thickening, truncation and distributed attenuation, suggesting colluvial wedge deposits close to the main fault zone (D ~ 1.5 m, inset b3). Vertical exaggeration is 2.

Figure 9: Results of the three-dimensional analysis and interpretation performed on the entire GPR dataset: a) 3D structural model of the Campotenese basin updated after Brozzetti et al., 2017a (DTM sources: TINITALY by Tarquini et al., 2012 and by Regione Calabria - www.regione.calabria.it, under license IODL 2.0. - https://www.dati.gov.it/iodl/2.0/); b) GPR section view (cmt1n-b) with interpretation including synthetic and antithetic fault splays (blue= W-to SW-dip; red=E-to NE-dip, respectively); c) detailed structural scratch of faults obtained by the analysis and correlation of interpreted fault slapys across the entire GPR dataset; the inset c1 is a conventional structural map oriented to the North and reporting the same fault sets to highlight the maximum width derived for the fault zone d) synthetic stereo-net plots of the fault planes in c), reporting the mean Dip Azimuth / Dip angle extracted for the identified four main sets of discontinuities, with a Dip Azimuth ranging between N 235-245° and N 062-072° for the W-dipping and E-dipping normal faults, respectively. Vertical exaggeration is 2.

Table 1: Main information and GPR parameters used during the data collection (* the time window was adapted depending on the surveyed area).

Table 2: Customized flow and details of the parameters used during the processing of the GPR dataset.